# Interrogating the degradation pathways of unstable mRNAs with XRN1-resistant sequences

Volker Boehm[1,*], Jennifer V. Gerbracht[1,*], Marie-Charlotte Marx[1] & Niels H. Gehring[1]

The turnover of messenger RNAs (mRNAs) is a key regulatory step of gene expression in eukaryotic cells. Due to the complexity of the mammalian degradation machinery, the contribution of decay factors to the directionality of mRNA decay is poorly understood. Here we characterize a molecular tool to interrogate mRNA turnover via the detection of XRN1-resistant decay fragments (xrFrag). Using nonsense-mediated mRNA decay (NMD) as a model pathway, we establish xrFrag analysis as a robust indicator of accelerated 5′–3′ mRNA decay. In tethering assays, monitoring xrFrag accumulation allows to distinguish decapping and endocleavage activities from deadenylation. Moreover, xrFrag analysis of mRNA degradation induced by miRNAs, AU-rich elements (AREs) as well as the 3′ UTRs of cytokine mRNAs reveals the contribution of 5′–3′ decay and endonucleolytic cleavage. Our work uncovers formerly unrecognized modes of mRNA turnover and establishes xrFrag as a powerful tool for RNA decay analyses.

[1] Institute for Genetics, Department of Biology, University of Cologne, Zuelpicher Straße 47a, 50674 Cologne, Germany. * These authors contributed equally to this work. Correspondence and requests for materials should be addressed to N.H.G. (email: ngehring@uni-koeln.de).

Degradation of messenger RNAs (mRNAs) directly influences the number of transcripts available for translation and thereby controls the persistence of the genetic information[1]. The regulation of mRNA degradation is widely used as a molecular principle underlying the correct expression of genes, for example of the immune system[2,3]. Many cytokine mRNAs exhibit short half-life times due to destabilizing sequences, which allow to fine-tune inflammatory responses[4,5]. In addition, the RNA decay machinery monitors the quality of mRNAs and eliminates faulty transcripts[2]. A well-studied example is the degradation of transcripts containing premature termination codons (PTCs) by the nonsense-mediated mRNA decay pathway (NMD)[6,7].

The process of mRNA decay is regulated by a large number of trans-acting protein factors. Many of them are mRNA-binding proteins, which directly interact with their target mRNAs via specific binding sites[8,9]. Destabilizing and stabilizing factors antagonistically regulate the turnover of bound mRNAs. Hence, the combination of RNA-binding proteins on a given mRNA will eventually determine the half-life of this mRNA and thereby its fate and potential for protein production.

Three major pathways of mRNA decay exist in mammalian cells: 5′–3′ exonucleolytic-, 3′–5′ exonucleolytic- and endonucleolytic decay. Deadenylation is considered to be the first and rate-limiting step during the turnover of normal cellular mRNAs[10]. Accelerated deadenylation is often initiated by the specific recruitment of the CCR4-NOT deadenylase complex, for example, by the SMG5–SMG7 heterodimer during NMD[11]. After deadenylation, further degradation of the mRNA occurs via the cytoplasmic Lsm1-7-Pat1 complex in combination with the eIF4E-binding protein 4E-T, which recruit decapping factors to the 5′ end of the deadenylated mRNA[12,13]. After decapping, the 5′–3′ exoribonuclease XRN1 recognizes the 5′ monophosphate and degrades the entire transcript[14,15]. Alternatively, the deadenylated mRNA may also be eliminated by the cytoplasmic exosome, a multi-protein complex, via its 3′–5′ exonucleolytic subunit DIS3L (ref. 16). Furthermore, transcripts with shortened poly(A) tails can be 3′-oligouridylated by TUTases, activating different degradation pathways including the 3′–5′ decay via the exosome-independent DIS3L2 (refs 17–19).

The decay of mRNA may also be initiated by endonucleolytic cleavage. A few endonucleases have been described in mammalian cells, amongst them ZC3H12A (Regnase-1) and the NMD-specific endonuclease SMG6, both of which harbour a PIN-like RNase domain[20]. Regnase-1 specifically recognizes and cleaves a stem–loop structure present in the 3′ UTRs of many cytokine mRNAs, such as TNF-α and Interleukin 6 (ref. 4). In contrast, SMG6 is recruited to PTC-containing mRNAs and cleaves them in the vicinity of the termination codon[21].

Despite the importance of mRNA turnover for the regulation of gene expression, little is known about the contribution of degradation pathways to the decay of individual mRNAs. This lack of knowledge is partially due to the large variety of seemingly redundant nucleolytic enzymes. In this study, we aimed to understand the degradation of different classes of intrinsically unstable mRNAs in mammalian cells using a virus-derived RNA sequence. Insertion of XRN1-resistant sequences (xrRNAs) into different reporter mRNAs allowed to monitor RNA decay activity of NMD substrates, cytokine 3′ UTR-containing transcripts and mRNAs containing AU-rich- or microRNAs (miRNA)-responsive elements. Using this method, we detect for these unstable transcripts a differential contribution of mRNA degradation pathways, including endocleavage, deadenylation and decapping.

## Results

### Monitoring NMD activity using XRN1-resistant RNA elements.

Decapping or endocleavage of mRNAs generates decay intermediates with 5′ unprotected ends, which are substrates of the cytoplasmic 5′–3′ exonuclease XRN1 (Fig. 1a). The depletion of XRN1 is commonly used to detect decay intermediates and to draw conclusions about the mRNA decay mechanism. However, XRN1 interacts directly with components of the decapping machinery, suggesting that XRN1 itself modulates the decapping process[22]. Hence, we sought to develop a system to monitor 5′–3′ mRNA decay with minimal cellular invasiveness. A potential alternative to the knockdown of XRN1 is the incorporation of viral xrRNA elements in reporter mRNAs (Fig. 1a), which were previously reported to block the processively degrading XRN1 upstream of the xrRNA structure[23]. The resulting XRN1-resistant decay fragments (henceforth called xrFrag) accumulate and can be used as a readout for mRNA degradation pathways involving 5′–3′ decay (Fig. 1a).

To test the feasibility of this approach, we inserted an xrRNA element from the Murray Valley encephalitis (MVE) Virus downstream of the triosephosphate isomerase (TPI) open reading frame (ORF; Fig. 1b,c). The MVE xrRNA is structurally and molecularly well described and consists of two xrRNA sequences (xrRNA1 and xrRNA2)[23]. By northern blot analysis of transiently transfected cells we observed in addition to the control mRNA two reporter-derived RNA species. The slower migrating band represents the full-length TPI-xrRNA transcript, whereas the weak, faster migrating band corresponds in size to the expected xrFrag and likely results from regular mRNA turnover (Fig. 1d). Introducing an NMD-activating PTC (PTC160) in the TPI ORF resulted in decreased steady state reporter levels as well as increased xrFrag abundance, in line with the enhanced turnover of this mRNA (Fig. 1d, lane 2). Interestingly, no xrFrags were detected from reporter mRNAs with single xrRNA sequences (xrRNA1 or xrRNA2), establishing that two xrRNA structures (constituting one complete xrRNA element) are required for efficient XRN1 resistance. To investigate the influence of upstream RNA elements on xrRNA functionality, we inserted 60 bp sequences with varying GC contents (30–70% GC) derived from the RAB7A 3′ UTR. To the best of our knowledge, this RNA does not contain decay-inducing features and is recommended as housekeeping gene for gene expression studies[24] (Fig. 1e). To simulate more challenging potential roadblocks for XRN1, we inserted 4MS2 binding sites or a very stable stem–loop structure upstream of the xrRNA (Fig. 1e). All inserts were fully compatible with xrFrag generation (Fig. 1f), thus establishing that the xrRNA element can be readily integrated into different reporter systems.

We next asked if the xrFrag could be used to study the degradation of NMD substrates in more detail. To this end we expressed wild type (WT) or PTC-containing versions of β-globin or TPI reporter mRNAs with or without xrRNA sequences (Fig. 2a and Supplementary Fig. 1a). The presence of the xrRNA element did not alter the overall degradation efficiency of the reporter (Fig. 2b and Supplementary Fig. 1b). Furthermore, an increase of xrFrag levels was consistently detected for both PTC-containing reporters. The currently assumed major degradation pathway of NMD substrates is the SMG6-catalysed endonucleolytic cleavage at the stop codon. Of the resulting unstable 5′- and 3′ fragments, the 3′ fragments are rapidly removed by XRN1 (refs 25,26). 3′ fragments were stabilized by the knockdown of XRN1 (Fig. 2c) and appeared in addition to the xrFrag (Fig. 2b and Supplementary Fig. 1b). Depletion of the key NMD factor UPF1 abolished the difference in xrFrag:reporter ratios for the WT and PTC reporters, indicating that NMD activity is required for the increased xrFrag levels. To gain more insight into the generation of the xrFrag, we established stable

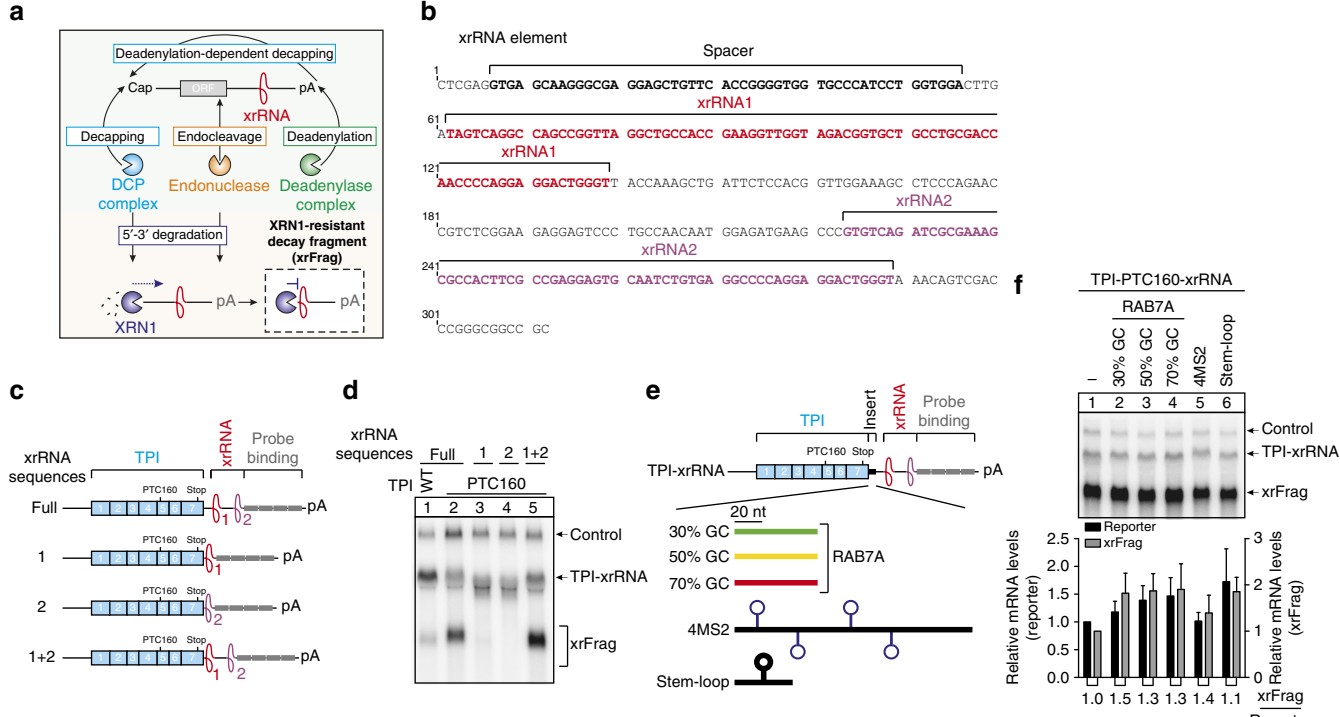

**Figure 1 | Characterization of xrRNA elements enabling the detection of mRNA degradation intermediates.** (**a**) Depicted are the general mRNA degradation pathways leading either directly (decapping and endocleavage) or indirectly (deadenylation) to 5′–3′ decay executed by XRN1. The presence of a stable XRN1-resistant RNA structure (xrRNA) prevents XRN1 from further progression and thus protects the remaining RNA fragment (xrFrag) from degradation from the 5′ end. (**b**) The DNA sequence of the xrRNA element used in reporter constructs is shown with annotations of sequence motifs. (**c,e**) Schematic representation of the TPI reporter mRNA. The TPI gene is depicted as blue boxes representing single exons (exon numbers indicated). The positions of the normal stop codons (stop) and premature translation termination codons are shown. Northern blot probe binding sites in the 3′ UTR are depicted as grey boxes and single xrRNA structures 1 and 2 are shown in red and purple. The difference between full xrRNA and 1 + 2 is the presence or absence of a short spacer region (indicated in **b**). (**e**) The 60 bp elements with varying GC content were derived from the RAB7A 3′ UTR, the 4MS2 binding sites are identical to those used in tethering experiments (Fig. 5) and the stem–loop structure is used in other reporters to block translation initiation (Fig. 5). (**d,f**) Northern blots of RNA samples extracted from HeLa cells transfected with the indicated reporter constructs. Co-transfected LacZ served as control mRNA. (**f**) Mean values of reporter and xrFrag signal ± s.d. (n = 3) were quantified and normalized to the TPI reporter without insert. The ratio of xrFrag to reporter mRNA levels is indicated below the graph.

tetracycline-inducible HeLa cell lines expressing β-globin or TPI reporter mRNAs with 3′ UTR xrRNA elements. Although we observed qualitatively similar results as with transient transfections, the overall effects were more pronounced when stable cell lines were analysed (Fig. 2d,e and Supplementary Fig. 1c,d). To determine the cellular degradation site of the PTC39 substrate, we isolated RNAs after sub-cellular fractionation. The full-length reporter was abundant in the nuclear fraction, but almost undetectable in the cytoplasm (Supplementary Fig. 1e). In contrast, xrFrags were preferentially found in the cytoplasm. These observations demonstrate that the xrFrags accumulate due to the degradation of the NMD substrate after the export to the cytosol.

We next examined the kinetics of the synthesis and decay of the reporter mRNAs by a time-course assay. Whereas the control WT mRNA increased in abundance during the first 8 h after induction, no PTC mRNA accumulated when UPF1 was present (Fig. 2f and Supplementary Fig. 1f). Interestingly, high xrFrag levels were detected already 4 h after induction. This indicates very rapid degradation kinetics of the PTC mRNAs, which was visualized via the concurrent detection of xrFrags. Inhibiting translation by cycloheximide treatment before induction abolished both, PTC reporter degradation and xrFrag generation (Supplementary Fig. 1g). Likewise, treating the cells with caffeine, a known inhibitor of the NMD-kinase SMG1 (ref. 27), resulted in

a dose-dependent increase of PTC-reporter abundance accompanied by a decreased accumulation of xrFrag (Supplementary Fig. 1h). Taken together, the application of xrRNA elements enables the simultaneous detection of both full-length reporter mRNAs and 5′ processed decay intermediates, which in combination greatly improves the readout of RNA degradation analyses.

Next, we monitored the stability of the xrFrag as well as the TPI–WT and two different NMD substrate mRNAs (TPI-PTC160 and TPI-SMG5; Fig. 3a) for 6 h after inhibition of transcription by actinomycin D. Compared with TPI–WT the degradation of both NMD substrates was accelerated, albeit with different kinetics (Fig. 3b). In contrast, the xrFrags generated from the NMD reporter mRNAs were more stable and exhibited a delay before decay commenced, leading overall to longer apparent half-life time (Fig. 3b). To further study the turnover of xrFrags, we chased the decay intermediates of globin-PTC39 with actinomycin D treatment in XRN1-depleted cells after a 4 h induction of transcription (Fig. 3c). Interestingly, the readily visible 3′ fragments exhibited a similar decay kinetic as the xrFrags of control cells. By contrast, in XRN1-depleted cells the xrFrags increased in abundance during the first 4 h after transcriptional shutoff. This suggests that residual XRN1 molecules will continue to degrade 3′ fragments, leading to a constant formation of xrFrags from the pool of 3′ fragments. We

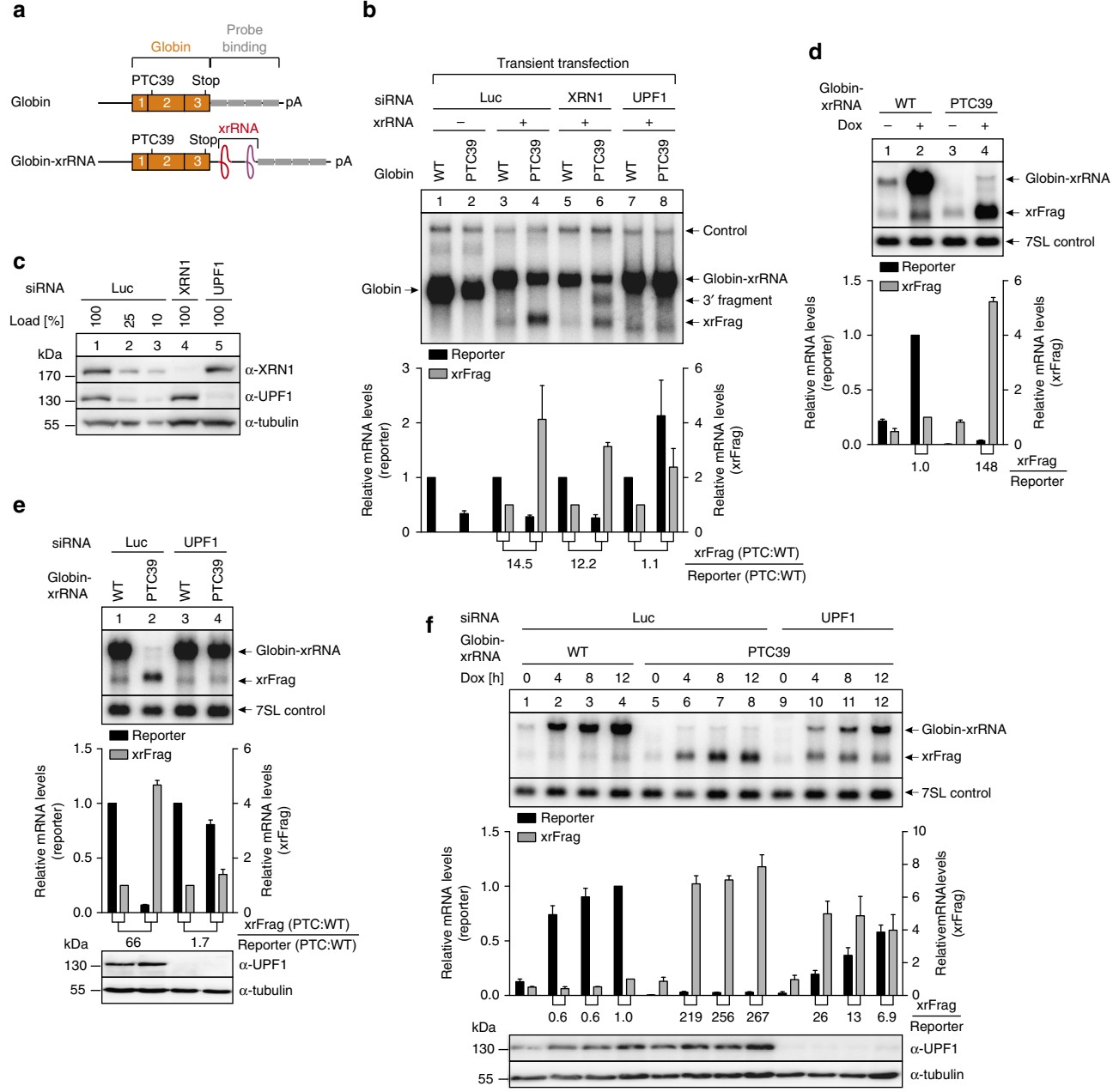

**Figure 2 | Degradation of NMD substrates is traceable by xrFrag analysis.** (**a**) Depiction of the β-globin reporter mRNAs as in Fig. 1. (**b**) Northern blot of RNA samples extracted from HeLa cells transfected with the indicated siRNAs and reporter constructs. Co-transfected LacZ served as control mRNA. Mean values of reporter and xrFrag signal ± s.d. ($n = 3$) were quantified and for each knockdown condition the PTC values were normalized to the WT. The ratio of xrFrag to reporter mRNA levels is indicated below the graph. (**c**) Western blot analysis of siRNA knockdown efficiency using the indicated antibodies. Tubulin served as loading control. (**d–f**) Total RNA was extracted from stable HeLa Flp-In T-REx cells expressing the indicated reporter RNA and analysed by northern blotting. (**e,f**) The cells were transfected with the indicated siRNA 72 h before induction of expression. 7SL RNA served as endogenous control RNA. Unless indicated otherwise (**f**), reporter mRNA expression was induced for 24 h with 1 μg ml$^{-1}$ doxycycline (Dox). Mean values of reporter and xrFrag signal ± s.d. ($n = 3$) were quantified and normalized to the WT control (+ Dox for **d**; Luc or UPF1 knockdown for **e**; 12 h after Dox for **f**). The ratio of xrFrag to reporter mRNA levels is indicated below the graph.

therefore hypothesized that in XRN1-depleted cells the production of 3′ fragments would precede the accumulation of xrFrags. Indeed, while 3′ fragments were more abundant than xrFrags in the first hours after transcriptional induction, this ratio decreased after 8 h of continuous PTC reporter expression (Fig. 3d). To determine whether xrFrags are degraded over time via continuous 5′–3′ decay, we constructed mRNA reporters containing additional xrRNA elements (that is, in total 2x- and 3x-xrRNA elements; Supplementary Fig. 2a). Interestingly, expression of

these reporters resulted in the appearance of additional xrFrags (Supplementary Fig. 2b,c). This finding suggests that in mammalian cells xrRNAs do not irreversibly trap, but impair XRN1 activity to such an extent that xrFrags can be detected.

**Dissecting pathways involved in NMD using xrFrag analysis.** In current models of NMD, phosphorylated and RNA-bound UPF1 serves as a docking platform for a variety of decay-inducing

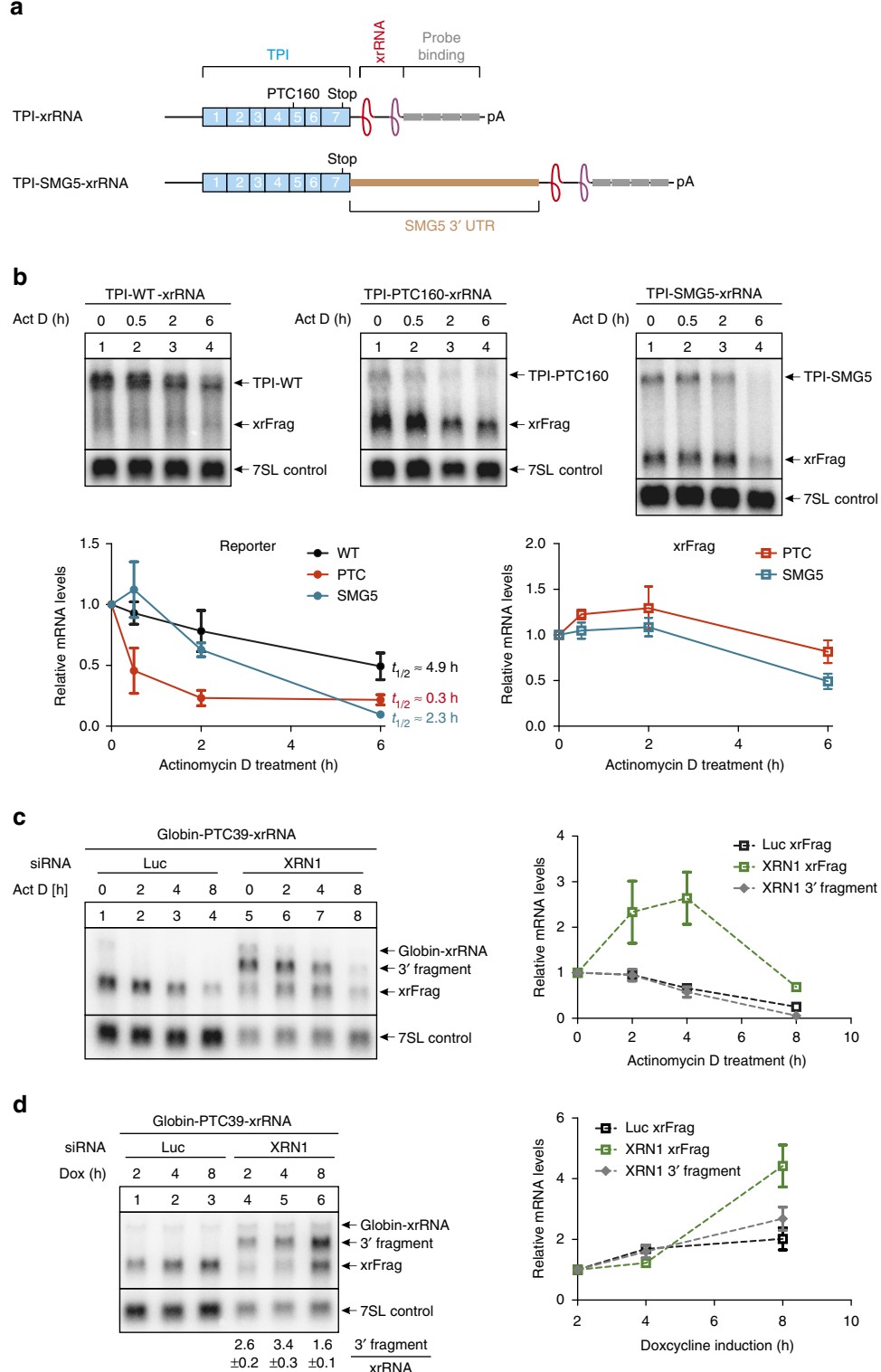

**Figure 3 | Kinetics of xrFrag accumulation and degradation.** (**a**) Schematic representation of the TPI reporter mRNA as in Fig. 1. The NMD-activating long 3′ UTR fragment of SMG5 is indicated. (**b–d**) Northern blot analysis of RNA samples derived from HeLa stable cell lines expressing the indicated reporter constructs as described in Fig. 2. Mean values of reporter, xrFrag and 3′ fragment signal ± s.d. ($n = 3$) were quantified and normalized to the 7SL endogenous control. (**b**) 2 h after induction of transcription by doxycycline (Dox), actinomycin D ($5\,\mu g\,ml^{-1}$) was added and the cells harvested at the indicated time points. (**c**) Transcription was induced for 4 h with Dox and reporter mRNAs were chased for the indicated time with actinomycin D. (**d**) Induction of reporter transcription was performed for the indicated time with Dox.

factors (Fig. 4a). We decided to determine if the xrRNA system is able to detect differences between redundant mRNA decay pathways that are activated during NMD. To this end, we

performed siRNA-mediated knockdowns of SMG6 and SMG7 (Fig. 4b). In line with published data[28], the depletion of either SMG6 or SMG7 did not completely stabilize PTC to WT mRNA

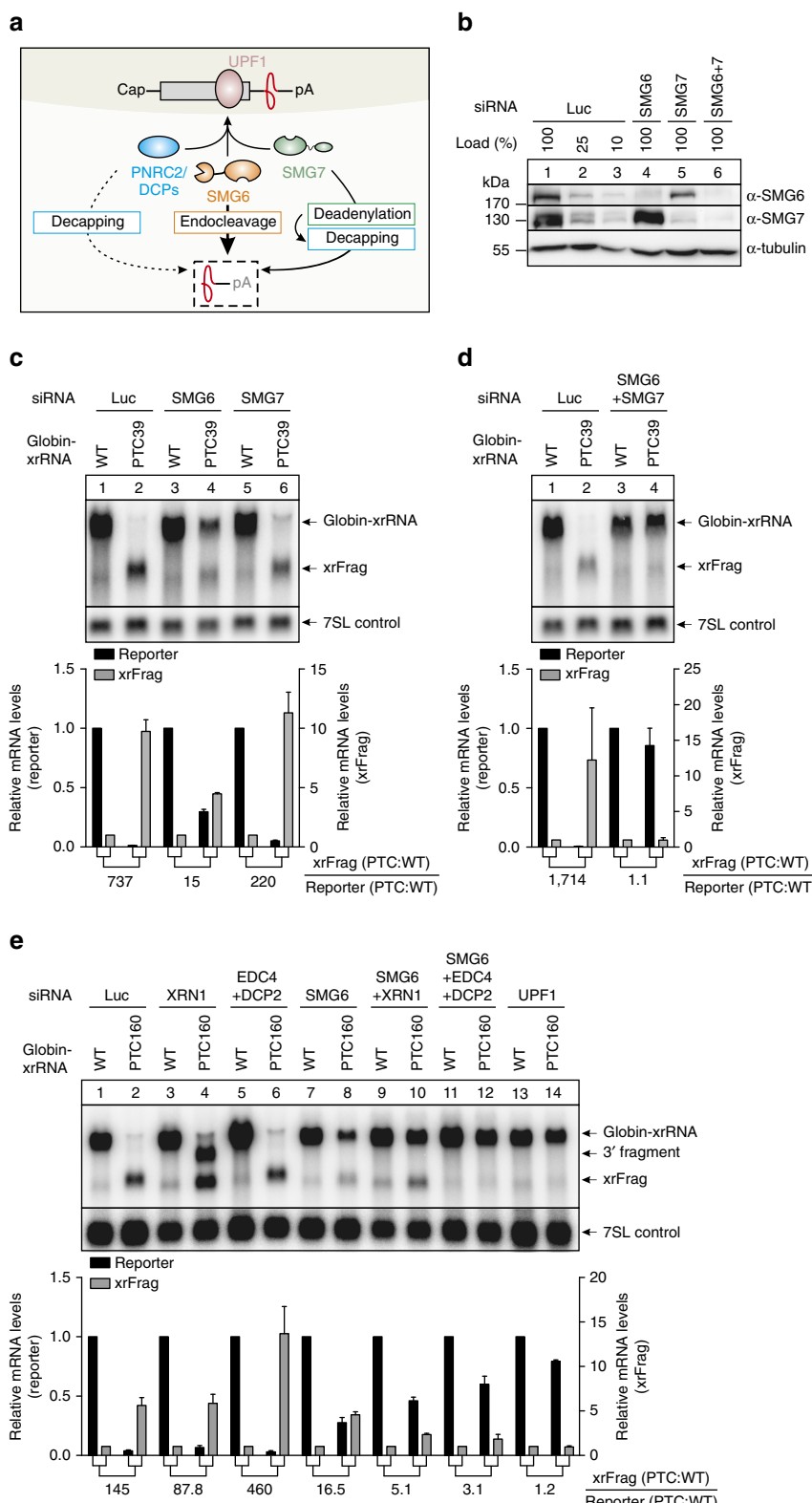

**Figure 4 | Tracing distinct degradation pathways employed during NMD by monitoring xrFrags.** (**a**) Model showing the recruitment of NMD-specific decay-inducing factors to activated UPF1 and their contribution to xrFrag accumulation. (**b**) Western blot analysis of knockdown efficiencies was performed with the indicated antibodies, tubulin served as loading control. (**c**–**e**) Northern blots of RNA samples extracted from stable HeLa cell lines transfected with the indicated siRNAs and expressing the indicated reporter constructs. Endogenous 7SL served as control RNA. Mean values of reporter and xrFrag signal ± s.d. ($n = 3$) were quantified and for each knockdown condition the PTC values were normalized to the WT. The ratio of xrFrag to reporter mRNA levels is indicated below the graph.

levels (Fig. 4c and Supplementary Fig. 3a). Nevertheless, we found that compared with SMG7 depletion, the knockdown of SMG6 decreased xrFrag:reporter ratios more profoundly for both β-globin and TPI reporter mRNAs. Furthermore, we detected almost identical xrFrag:reporter levels for WT and PTC reporter upon the combined SMG6–SMG7 knockdown, indicating near-complete NMD inhibition (Fig. 4d and Supplementary Fig. 3b). Taken together, these results are in line with the current NMD model, in which SMG6 is the major and SMG7 the alternative degradation-promoting factor.

We next aimed to distinguish decapping from endocleavage activities during NMD. To this end, we generated a reporter with a 5′ xrRNA (xrA). We inserted a 5′ stem–loop and a NanoLuc ORF upstream of the xrA to prevent ribosomes from disrupting the xrRNA structure. Downstream of the xrA, an EMCV IRES followed by the TPI ORF with or without a second xrRNA (xrB) was cloned (Supplementary Fig. 4a). The activation of NMD by IRES-mediated translation will either lead to endocleavage or decapping and the accumulation of xrFragB or xrFragA, respectively. We found decreased levels of the PTC-containing mRNA and increased xrFragB compared with the WT. Surprisingly, strongly reduced amounts of xrFragA were detected, too (Supplementary Fig. 4b). This finding underlines that as long as SMG6 is active in the NMD pathway, endocleavage dominates over decapping. Indeed, increased levels of PTC-containing xrFragA accumulated in SMG6-depleted cells, leading to identical xrFrag:reporter ratios of the PTC and the WT construct (Supplementary Fig. 4c).

If deadenylation-dependent decapping occurs during NMD, we expect that decapped full-length reporter mRNAs appear when the decapping machinery or XRN1 are depleted together with SMG6. Although we readily detect 3′ fragments as a result of endocleavage, knockdown of XRN1 alone lead to only marginally increased levels of the full-length reporter RNA (Fig. 4e and Supplementary Fig. 3c,d). To shut down the decapping machinery, we used siRNAs targeting the catalytic subunit DCP2 and the scaffold protein EDC4 (also called Hedls)[29]. We observed no accumulation of PTC reporter mRNA upon EDC4 + DCP2 depletion. When SMG6 and XRN1 were depleted together, no 3′ fragments were detected and reporter levels increased twofold compared with SMG6 knockdown alone, suggesting that 5′–3′ decay becomes more important when SMG6 is depleted. Surprisingly, also the combined SMG6 + EDC4 + DCP2 knockdown could not fully increase the PTC reporter to WT levels, even though elevated PTC reporter levels were observed. Interestingly, the knockdown of UPF1 increased the reporter RNAs more than the combined SMG6 + XRN1 or SMG6 + EDC4 + DCP2 depletion, suggesting that not all degradation during NMD can be attributed to decapping or endocleavage. However, we cannot exclude the possibility that the incomplete depletion of the decay factors contributes to the observed discrepancy. Taken together, xrFrag analysis suggests that 3′–5′ degradation pathways contribute to the removal of NMD substrates, when 5′–3′ decay is inhibited.

**5′–3′ decay is initiated by decapping and endocleavage**. Having investigated the decay routes of the complex NMD pathway in detail, we next used the MS2 tethering system to study individual mRNA-degrading proteins by recruiting them to an xrRNA-containing reporter mRNA (Fig. 5a).

Tethering of the DCP1 interacting and decapping-promoting protein PNRC2 reduced the levels of the TPI reporter mRNA and increased xrFrag levels more than twofold (Fig. 5b). The W114A mutant of PRNC2, which is unable to bind DCP1 (ref. 30), was inactive in the tethering assay. However, tethering

of the ΔC mutant, which lacks the C-terminal NR box required for the UPF1 interaction[30], showed a degradation similar to the WT. In contrast to the decapping factor PNRC2, the NMD factor SMG7 recruits via its C-terminus the catalytic subunit POP2 of the CCR4-NOT complex leading to mRNA deadenylation[11]. The accelerated deadenylation is followed by decapping and degradation from the 5′ end[31]. Interestingly, when we tethered SMG7 full-length or only the C-terminal domain, which both lead to reduced TPI reporter mRNA, we detected slightly decreased rather than increased xrFrag levels (Fig. 5c). We therefore wanted to compare the impact of decapping, endocleavage or deadenylation on the generation of xrFrag. To this end we tethered the C-terminal domain of SMG7 (deadenylation), PNRC2 (decapping) and the EJC core component BTZ, which was shown to activate endocleavage via NMD[26]. Interestingly, tethering of PNRC2 and BTZ resulted in decreased reporter levels and increased levels of xrFrag, whereas SMG7 decreased the amounts of the reporter as well as xrFrag (Fig. 5d). The effects of tethered SMG7 and PNRC2 were also observed with a reporter (SL-NL-4MS2-xrRNA), which is inefficiently translated due to a strong stem–loop in the 5′ UTR (Fig. 5e–g). Hence, both proteins act in a translation-independent manner. Because NMD is a translation-dependent process, BTZ was inactive when tethered to the non-translatable SL-NL-4MS2-xrRNA mRNA. Our results suggest that SMG7-mediated degradation does not only enhance 5′–3′ decay, but also leads to the degradation of the substrate mRNA by 3′–5′ decay. This could also explain the remaining degradational activity we observed in the analysis of NMD substrates after combined knockdowns of SMG6 with XRN1 or decapping factors (Fig. 4e and Supplementary Fig. 3c). To test this idea, we constructed a tethering reporter containing a region from the MALAT1 lncRNA, which is cleaved by RNAse P and leads to the formation of a stable 3′ triple helix (Supplementary Fig. 5a). This structure was shown to stabilize mRNA, likely by preventing the 3′–5′ decay machinery to engage with the protected RNA 3′ end[32]. Tethering of PNRC2 leads to equal degradation of polyadenylated or triple-helix-containing RNA (tH), whereas SMG7 preferentially degraded the polyadenylated reporter mRNA (Supplementary Fig. 5b,c).

We next asked if the tethering of other deadenylation-promoting factors would lead to mRNA degradation with reduced xrFrag accumulation. For these experiments, we chose the silencing domains of TNRC6A and TNRC6B, which are involved in the miRNA-induced mRNA decay pathway, as well as TTP, which mediates the degradation of AU-rich elements (AREs)-containing mRNAs. These proteins were shown before to induce degradation by recruitment of deadenylase complexes, thereby leading to accelerated deadenylation[33–35]. In our tethering assay, we could confirm that all tested deadenylation-promoting proteins resulted in reporter degradation, however, no increase in xrFrag was detected for any of these factors (Fig. 5h). To further delineate the contribution of deadenylation to the observed decay, we made use of a reporter in which the xrRNA was located 5′ of the MS2 binding sites (Supplementary Fig. 5d). When SMG7-C, TNRC6A-SD or TNRC6B-SD were tethered, the resulting xrFrags showed a faster migration (Supplementary Fig. 5e), which was visible in lane profiles for the respective xrFrag signals (Supplementary Fig. 5f). Taken together, we find a differential enhancement of 5′–3′ degradation when comparing decapping-, endocleavage- and deadenylation-promoting factors. The xrRNA analysis therefore represents an ideal starting point to characterize whether a given protein induces 5′–3′ mRNA decay.

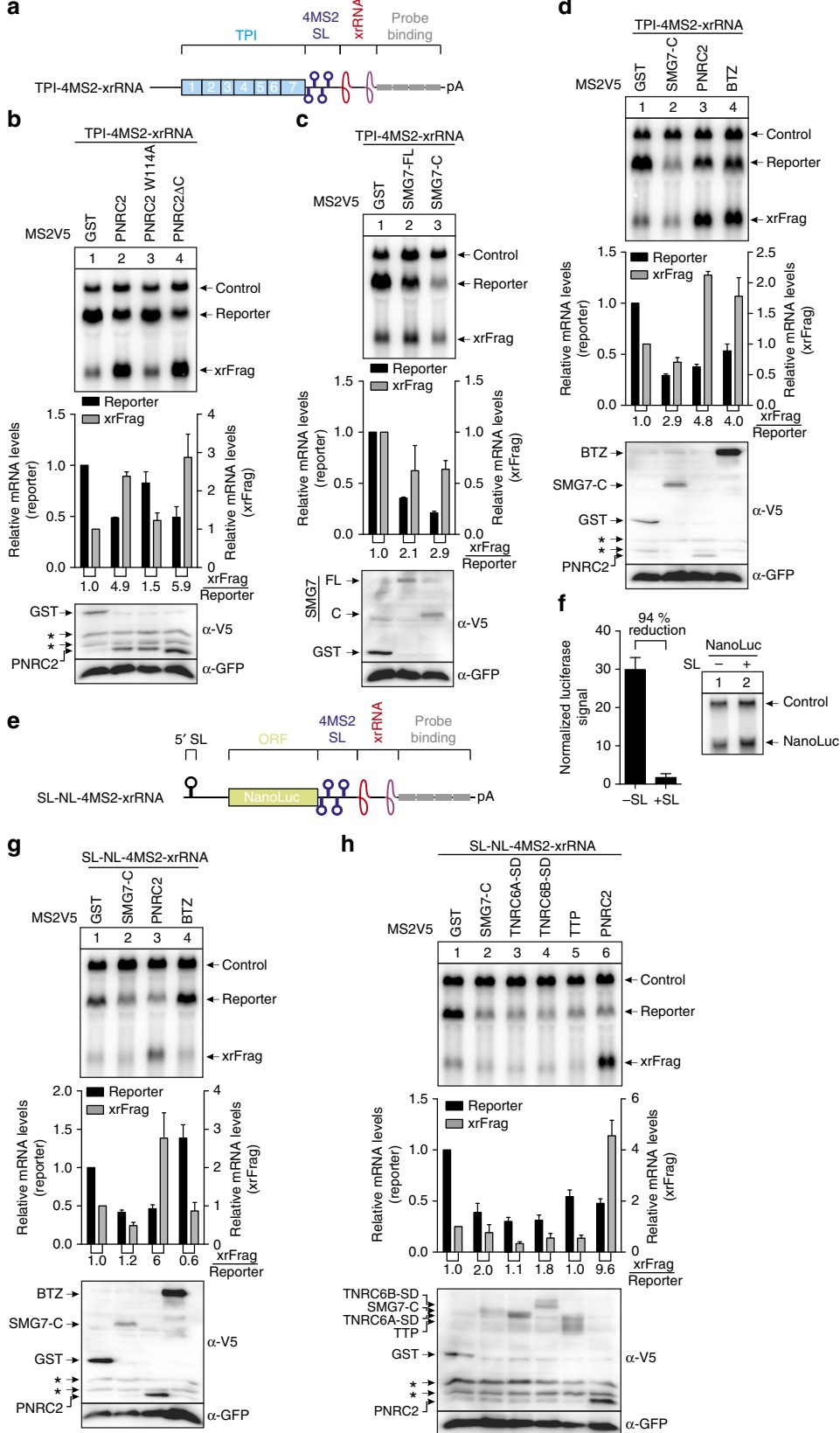

**Figure 5 | Characterization of mRNA decay induced by direct protein tethering via xrFrag analysis.** (**a**,**e**) TPI and NanoLuc tethering reporter mRNAs with 4MS2 binding sites are depicted as in Fig. 1. (**e**) The position of the 5′ stem–loop inhibiting ribosome scanning is indicated. (**b–d**, **g,h**) Northern blots of RNA samples extracted from HeLa cells transfected with the indicated tethering and reporter constructs. Mean values ± s.d. (*n* = 3) for reporter and xrFrag levels were quantified and normalized to tethered GST, which served as control. The ratio of xrFrag to reporter mRNA levels is indicated below the graph. Western blots show the expression levels of the MS2V5-tagged constructs with GFP serving as transfection control. Unspecific bands are indicated with asterisks. (**f**) Translational efficiency (mean ± s.d., *n* = 3) was measured by dual luciferase assay and compared for NanoLuc reporter with or without the 5′ stem–loop. Expression of the NanoLuc mRNAs is shown by northern blotting, co-transfected LacZ served as control.

**Characterization of isolated deadenylation-inducing elements.** So far, the xrFrag analysis allowed the dissection of complex decay pathways (NMD), as well as the characterization of degradation-inducing proteins (tethering). Next, we wanted to use xrRNAs to examine the turnover of mRNAs with individual decay elements, for example ARE and microRNA response elements (miRE). AREs are mRNA-destabilizing sequences that are known to work in heterologous reporter mRNAs[36]. In our constructs the c-fos or the granulocyte–macrophage colony-stimulating factor (GM–CSF) ARE were followed by a single xrRNA element (Fig. 6a). Upon induction of stable cell lines, we observed increased amounts of xrFrag for both AREs, but only the GM–CSF ARE also strongly reduced the expression of the reporter mRNA (Fig. 6b). However, calculating the xrFrag:reporter ratios indicated that both AREs enhanced 5′–3′ decay. The decay mediated by the GM–CSF ARE occurred via deadenylation-dependent decapping, because the reduced expression as well as increased xrFrag abundance was abolished by the depletion of either XRN1 or the CCR4-NOT scaffold protein CNOT1 (Fig. 6c,d).

The expression of many mammalian genes is regulated by miRNAs, which assemble an RNA-induced silencing complex onto their target mRNAs and thereby induce their degradation. We inserted into our reporter two previously described miRE (ref. 37) containing canonical let-7 and miR-21 binding sites (Fig. 6e). A reporter with mutated miREs was used as negative control. In comparison to the miRE (Mut) mRNA, the expression levels of the miRE (WT) mRNA upon induction of stable cell lines were strongly reduced (Fig. 6f). Interestingly, the xrFrag:reporter ratio was just increased by a factor of ∼3, indicating that only some of the degradation occurred by 5′–3′ decay. Depletion of CNOT1 restored normal levels of reporter mRNA and xrFrag, confirming that miRE-mediated mRNA decay involves deadenylation by the CCR4-NOT complex (Fig. 6g). Our previous observation of deadenylation induced by TNRC6 tethering further supports this finding (Fig. 5h). In conclusion, xrRNA and xrFrag analysis can be used to study the characteristics of short mRNA-destabilizing sequence elements.

**Endonucleolytic cleavage within TNF-α and IL6 3′ UTRs.** Accelerated turnover of cytokine mRNAs represents a crucial regulation mechanism of immune responses, which is reported to utilize multiple decay elements and factors. To study this

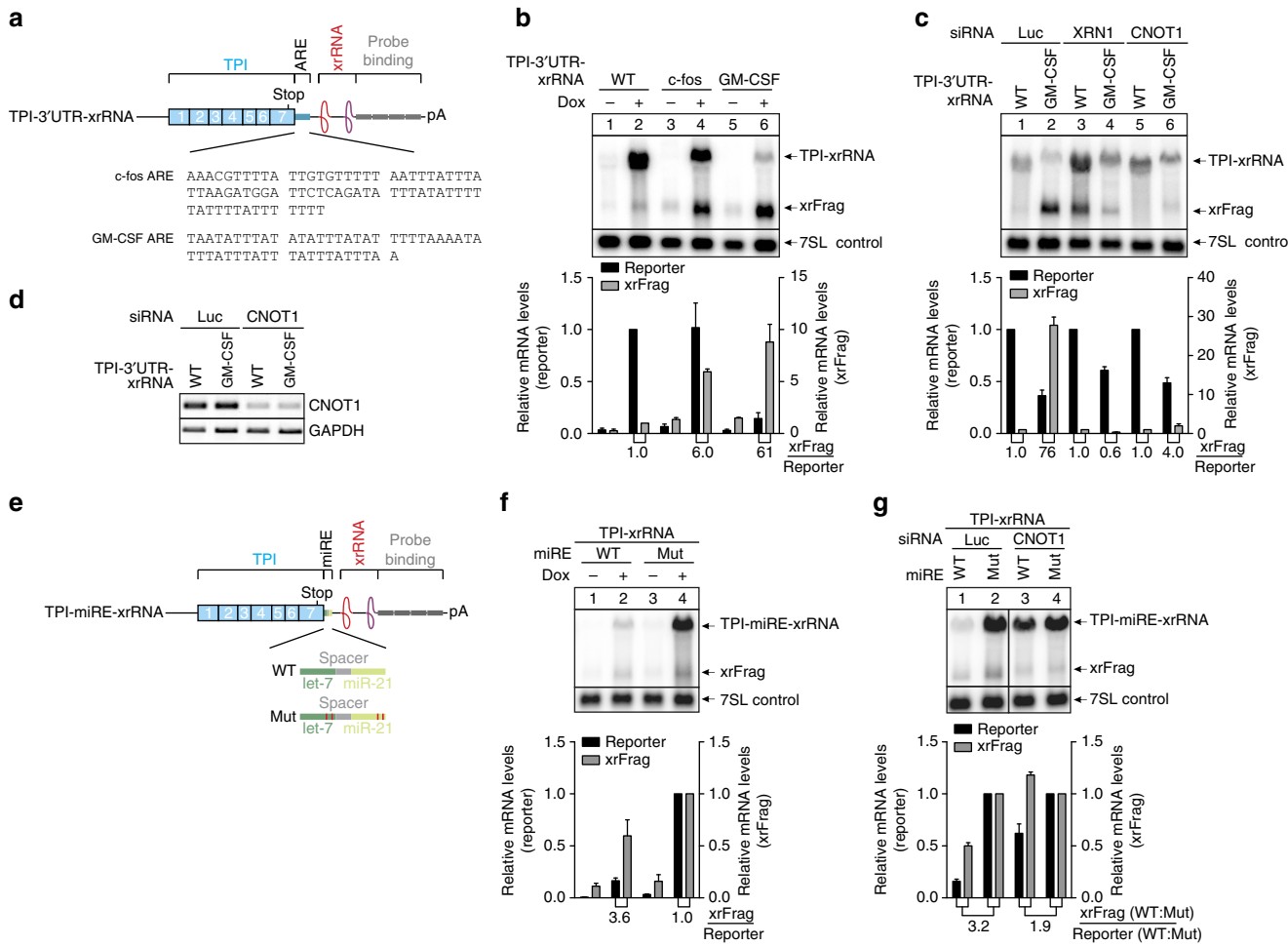

**Figure 6 | Isolated ARE and miRNA decay elements induce deadenylation-dependent mRNA degradation. (a,e)** Schematic representation of TPI reporter mRNAs depicted as in Fig. 1, containing (**a**) the decay-inducing AREs from c-fos and GM–CSF in the 3′ UTR or (**e**) WT or mutated miRE for let-7 and miR-21. (**b,c, f,g**) Hela Flp-In T-REx cells expressing the indicated constructs were harvested and RNA was extracted and analysed by northern blotting. (**c,g**) Knockdown was performed by transfecting the cells with the indicated siRNAs. Expression of reporter mRNAs was induced with 1 μg ml$^{-1}$ doxycycline (+ Dox) for 24 h. Endogenous 7SL RNA levels are shown as a control. Mean values of reporter and xrFrag signal ± s.d. (n = 3) were quantified and normalized to the WT control (+ Dox for **b** and **f**; per knockdown condition for **c** and **g**). The ratio of xrFrag to reporter mRNA levels is indicated below the graph. (**d**) Semi-quantitative PCR analysis of the CNOT1 and GAPDH (control) expression levels in siRNA treated cells.

degradation pathway using our xrRNA system, we inserted the ARE- and conserved stem–loop-containing regions of the TNF-α and IL6 3′ UTRs between the TPI ORF and the xrRNA element (Fig. 7a and Supplementary Fig. 6a,b). As a negative control we used a similar sized fragment of the RAB7A 3′ UTR (ref. 24). For both reporters (TNF-α and IL6) we observed in transiently transfected cells a strong reduction of full-length mRNA levels, which was accompanied by an increase in xrFrag abundance (Fig. 7b). Hence, TNF-α and IL6 3′ UTRs induce mRNA degradation that involves 5′–3′ decay. In stable cell lines, the TNF-α and IL6 full-length transcripts were barely detectable 24 h after induction, whereas the xrFrag levels were strongly increased compared with the RAB7A control (Fig. 7c). In time-course assays the RAB7A-containing mRNA increased in expression for 8 h after induction. In contrast, the xrFrag generated from the degradation of the TNF-α and IL6 reporter mRNAs reached maximum levels as fast as 4 h after the addition of doxycycline (Fig. 7d). These fast decay kinetics are similar to those observed in PTC-containing TPI and globin mRNAs and suggest that TNF-α and IL6 reporter RNAs are very efficiently degraded. Treating the cells with the translation inhibitors cycloheximide or puromycin before induction caused a strong increase of both TNF-α- and IL6-containing reporter mRNAs and almost indistinguishable xrFrag:reporter ratios compared with the RAB7A control (Fig. 7e,f). Hence, the majority of the degradation involves translation-dependent mechanisms.

TNF-α as well as IL6 mRNAs have been reported to be degraded by Regnase-1-mediated endonucleolytic cleavage[4]. In addition, both 3′ UTRs contain several AREs, which can lead to deadenylation followed by decapping. We reasoned that the insertion of two xrRNA elements flanking the 3′ UTR of a reporter mRNA would allow to discriminate mRNAs undergoing decapping and endocleavage. To this end, we constructed reporter mRNAs with two xrRNA elements, one upstream (xrA) and one downstream (xrB) of the 3′ UTR of interest (Fig. 8a). Expression of the xrA-RAB7A-xrB control reporter leads to the production of both xrFrag species with roughly equal levels (Supplementary Fig. 7a). For the xrA-TNF-α-xrB reporter, the levels of the longer xrFragA were reduced, whereas the shorter xrFragB accumulated in a translation-dependent manner (Supplementary Fig. 7a,b). The production of xrFragB was very fast and occurred during the first 4 h after induction (Supplementary Fig. 7c).

If both deadenylation-dependent decapping and endocleavage decay routes are used for the degradation, decapped full-length transcripts as well as 3′ fragments are expected to accumulate in cells lacking XRN1. Surprisingly, we did not detect increased amounts of decapped, full-length TNF-α and IL6 reporters in XRN1-depleted cells (Fig. 8b,c). Instead, we observed additional bands representing 3′ fragments, indicating that the enhanced 5′–3′ decay involves endonucleolytic cleavage. A conserved stem–loop in the TNF-α and IL6 3′ UTRs has been reported as binding site of Regnase-1 (ref. 4). We therefore asked, if the observed 3′ fragments are generated by endocleavage close to the stem–loop sequence. To this end, we amplified, cloned and sequenced the 3′ fragments of IL6 and TNF-α. Surprisingly, the cleavage sites of both reporters were outside the stem–loop sequence, 1–20 nucleotides upstream of the IL6 stem–loop and 132–138 nucleotides downstream of the TNF-α stem–loop (Supplementary Fig. 6a,b). To further analyse the endonucleolytic cleavage of these transcripts, we generated constructs in which the observed cleavage sites were deleted (Fig. 8d). Interestingly, the deletion of the mapped cleavage site resulted in a weak (TNF-α) or moderate (IL6) increase of the reporter mRNAs and reduced levels of xrFrag (Fig. 8e). In contrast, the slightly lower expression levels (compared with RAB7A) of a reporter mRNA

containing the isolated TNF-α cleavage site (Fig. 8d) resulted in mildly increased xrFrag accumulation (Fig. 8e). Upon XRN1 knockdown, TNF-α and IL6 3′ fragment levels were strongly reduced when the cleavage sites were deleted (Fig. 8f). Strikingly, in the TNF-α construct which contained the cleavage site, but not the stem–loop itself, we detected a 3′ endocleavage fragment. We mapped the cleavage site of this mRNA as described above to the same area as in the full-length TNF-α 3′ UTR (Supplementary Fig. 6a).

In conclusion, the xrFrag analysis helped to identify that TNF-α and IL6 3′ UTRs undergo endocleavage at previously uncharacterized positions outside the canonical Regnase-1 binding site. However, endocleavage can only partially explain the rapid degradation of both reporters. Hence, cytokine degradation also utilizes multiple major decay pathways for robust mRNA degradation, resembling the complex degradation mechanism observed for NMD substrates.

## Discussion

The analysis of mRNA decay intermediates has greatly facilitated the understanding of mRNA turnover mechanisms. Previously, a poly(G) tract was used in yeast to inhibit exonucleases and to study the directionality of mRNA degradation pathways[38,39]. However, poly(G) tracts fail to trap mRNA turnover intermediates in mammalian cells. In this work, we establish a viral-derived XRN1-resistant RNA sequence (xrRNA) as a molecular tool to study mRNA decay in mammalian cells. The insertion of xrRNA into different reporter constructs results in the robust and reproducible accumulation of degradation intermediates (xrFrag). Since the xrFrag is produced during the decay of the xrRNA-bearing mRNA, it enables to unambiguously discriminate mRNA decay from other causes of reduced mRNA levels. Moreover, the use of xrRNA to visualize decay intermediates provides insights into the contribution of 5′–3′ decay pathways. Amongst its advantages are the minimal interference with other cellular processes, simplicity of its application and versatility. Although high levels of xrFrags could potentially inhibit cellular 5′–3′ decay due to the sequestration or inactivation of XRN1, as suggested previously[40] we did not observe signs of such an inhibition. In fact, we detect only a single xrFrag band on our northern blots, whereas the primary 3′ fragments were detected as an additional band exclusively in cells lacking XRN1.

Although the xrFrag method generates robust data in our hands, it also offers the possibility for future improvements. It will be interesting to explore if any xrRNA sequences can be identified or evolved that inhibit XRN1 to a greater extent than the MVE virus xrRNA used in this work. We provide evidence that the MVE-derived xrFrags represent meta-stable intermediates of 5′–3′ decay, which will be eventually degraded. It remains to be determined if more stable xrFrags could allow even deeper insights into the molecular mechanisms of mRNA degradation. Interestingly, for exact xrFrag analysis a single xrRNA is superior to a combination of two or three xrRNAs, which do not significantly elevate the amount of the 5′ xrFrag, but lead to the accumulation of additional xrFrags. This seemingly paradoxical observation is explained by the slow, but constant conversion of longer xrFrags into shorter xrFrags.

Many of the analyses presented in this work employ the xrFrag method to monitor the contribution of 5′–3′ decay pathways to mRNA turnover. However, it is conceivable that xrFrag analysis may also be used to discriminate deadenylation-dependent and -independent modes of decay. Due to the small size of the xrFrags, their running behaviour is highly dependent on their poly(A) tail lengths. In addition, the defined 5′ end of the xrFrags offer superior resolution compared with the variable 5′ ends observed

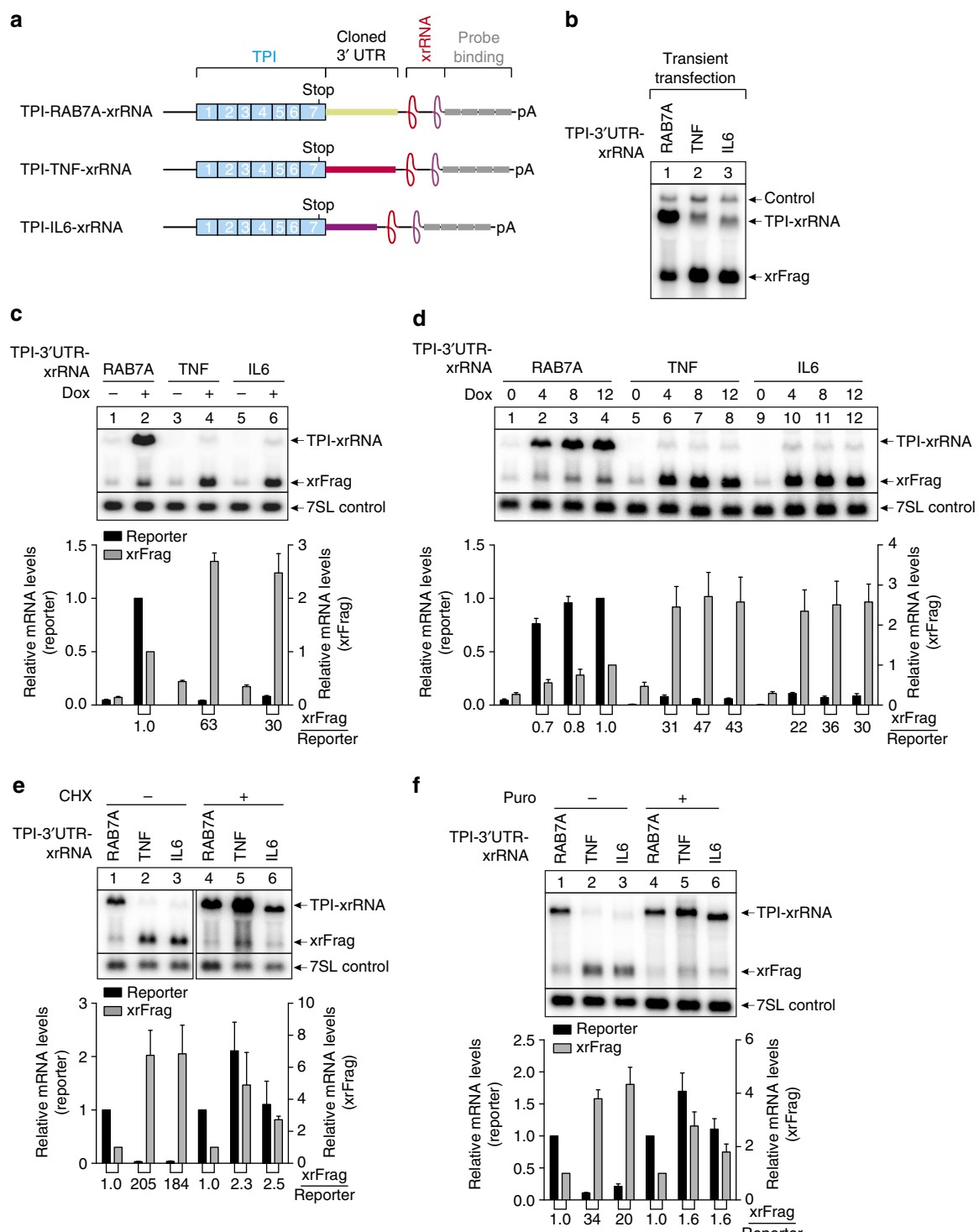

**Figure 7 | The degradation induced by TNF-α and IL6 3′ UTRs is translation dependent.** (**a**) TPI reporter mRNAs with RAB7A (control), TNF-α or IL6 (decay-inducing) 3′ UTRs are represented as in Fig. 1, with the inserted sequences shown as coloured boxes. (**b**) Northern blot analysis of RNA samples derived from HeLa cells transiently transfected with the indicated reporter constructs and LacZ as control. (**c–f**) HeLa Flp-In T-REx cells expressing the indicated reporter RNA were harvested, total RNA extracted and analysed by northern blotting. 7SL RNA serves as endogenous control RNA. Unless indicated otherwise (**c,d**), reporter mRNA expression was induced for 24 h with 1 µg ml$^{-1}$ doxycycline (Dox). Cycloheximide treatment together with doxycycline induction was performed for 8 h with 100 µg ml$^{-1}$ of cycloheximide. Puromycin (Puro) treatment together with doxycycline induction was performed for 4 h with 20 µg ml$^{-1}$ of puromycin. Mean values of reporter and xrFrag signal ± s.d. ($n = 3$) were quantified and normalized to RAB7A control ( + Dox for **c**; 12 h after Dox for **d**; with or without cycloheximide for **e**; with or without Puro for **f**). The ratio of xrFrag to reporter mRNA levels is indicated below the graph.

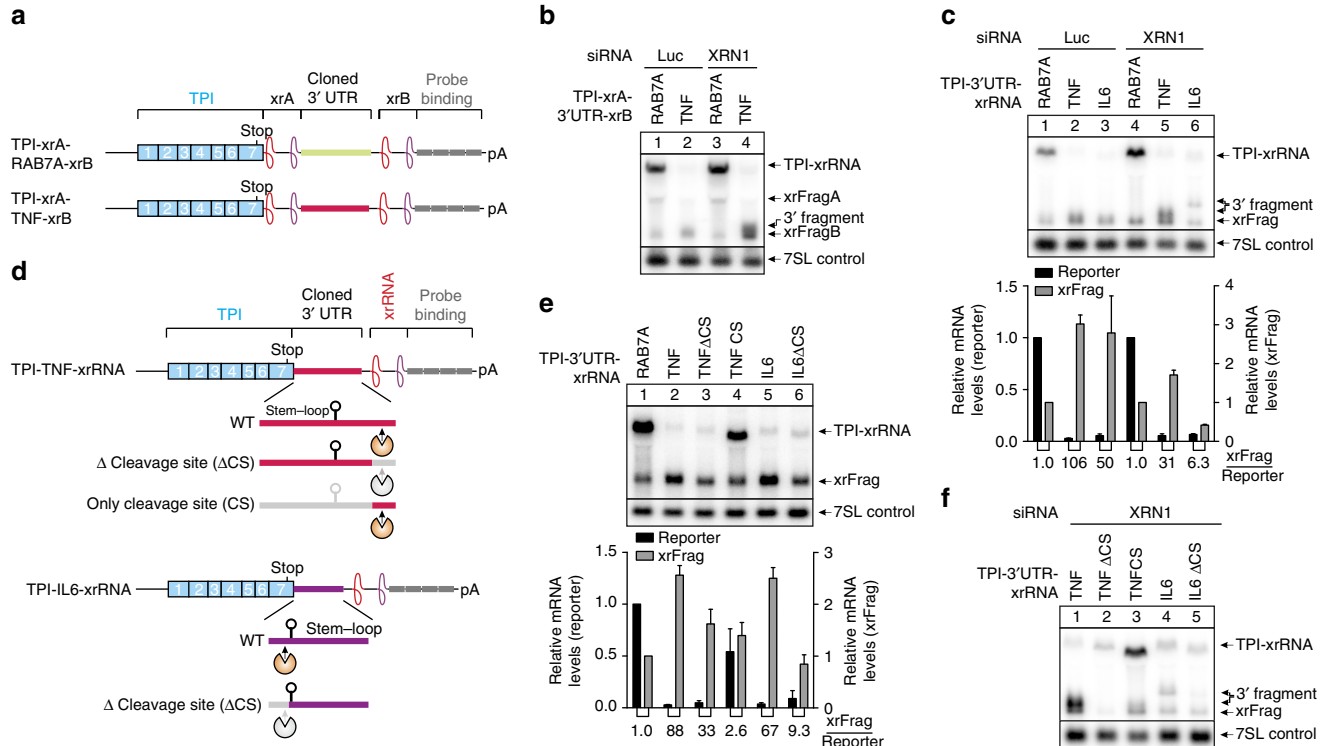

**Figure 8 | The degradation induced by TNF-α and IL6 3′ UTRs involves endocleavage.** (**a,d**) TPI reporter mRNAs with xrRNA-framed RAB7A (control), TNF-α or IL6 (decay-inducing) 3′ UTRs are represented as in Fig. 1, with the inserted sequences shown as coloured boxes. Dual xrRNA elements (xrA and xrB) are indicated. (**d**) Deletion mutants are shown containing the stem–loop, marked as a hairpin, or the cleavage site, marked by a schematic endonuclease. (**b,c**; **e,f**) HeLa Flp-In T-REx cells expressing the indicated reporter RNA were harvested, total RNA extracted and analysed by northern blotting. 7SL RNA serves as endogenous control RNA. The reporter mRNA expression was induced for 24 h with 1 µg ml⁻¹ doxycycline (Dox). Knockdown was performed by transfecting the indicated siRNA. (**c,e**) Mean values of reporter and xrFrag signal ± s.d. (n = 3) were quantified and normalized to RAB7A control. The ratio of xrFrag to reporter mRNA levels is indicated below the graph.

for 3′ fragments that accumulate upon XRN1 depletion. Although some of our results indicate that xrFrags allow to assess poly(A) tail lengths, specific assays such as RNaseH-oligo(dT) digestion or PAT assays[41] are needed to improve this aspect of our experimental system.

The xrRNA realized its full potential when used in combination with other approaches and proved to be fully compatible with mRNA half-life measurements and the knockdown of specific mRNA decay factors. This allowed to dissect the contribution of 5′–3′ decay pathways to the overall degradation of an mRNA. Furthermore, assessing the pharmacological inhibition of mRNA decay is very robust with the xrRNA reporter as shown by our analysis of cycloheximide-, puromycin- and caffeine-treated cells. Finally, we were able to establish NMD reporters with a previously unprecedented dynamic range between WT and PTC-containing mRNAs. Standard NMD analyses with transiently transfected reporter constructs often show 5–10-fold differences between the WT and the PTC-containing steady-state mRNAs. Our system on the one hand displays a 10–50 fold reduction of the expression of the NMD substrate compared with its WT counterpart. In addition, the differential accumulation of xrFrag leads to a dynamic range of up to three orders of magnitude when the ratio of full-length reporter and xrFrag is calculated. Furthermore, the knockdown of the central NMD factor UPF1 increases the expression level of the PTC-containing mRNA to the level of the WT mRNA. To the best of our knowledge, this quantitatively complete inhibition of NMD has not been accomplished with any other assay to date. The performance of our NMD assay is even more

remarkable in view of the possibility to follow the accumulation as well as the decay of NMD reporter and xrFrag over time. The degradation of the substrate mRNA occurs in a very short time frame and confirms that NMD eliminates bona fide substrate mRNAs with very rapid kinetics. In contrast, when XRN1 is abundant the xrFrags decay at a slower rate, which is similar to the one observed for 3′ fragments in XRN1-depleted cells. The comparable turnover rate of 3′ fragments and xrFrags is striking, given that their accumulation is caused by fundamentally different mechanisms of XRN1 inhibition. Importantly, we find that 3′ fragments are converted into xrFrags over time, which explains why xrFrags were still detected in XRN1-depleted cells.

The xrFrag analysis is also compatible with the tethering of RNA decay factors, which allows to disentangle complex mRNA turnover pathways, such as NMD. We show that the direct tethering of SMG7 to two different reporter mRNAs (translation-dependent and -independent) reduced the levels of the reporter mRNA, without substantial increase in the xrFrag:reporter ratio. The effect of tethered SMG7 was different from that of the tethered decapping activator PNRC2 or the tethered EJC-component BTZ, which both increased the amounts of xrFrags. Furthermore, SMG7 was less active in degrading a reporter mRNA containing the 3′ end of the MALAT1 lncRNA, which impairs 3′-5′ decay[32,42]. Our data are not only in line with the previously reported contribution of 3′–5′ decay to the overall activity of NMD[43], but also demonstrates the successful combination of xrFrag analysis with well-established tethering assays.

Our analysis of AREs and miREs demonstrates that xrRNAs can also be used to study individual decay-inducing sequences. We find that both degradation elements require deadenylation activity by the CCR4-NOT1 complex, however only for AREs we observed a substantial increase in 5′–3′ decay. Combined with the results of tethered deadenylation-promoting factors, these findings support the notion that shortening of the poly(A) tails is not exclusively followed by enhanced decapping.

We also analysed mRNA turnover that is mediated by two well-studied cytokine 3′ UTRs, TNF-α and IL6. Different destabilizing sequences have been identified within these 3′ UTRs, for example, AREs. Furthermore, both 3′ UTRs contain a stem–loop structure, which is recognized by the RNA-decay factors Regnase-1 and Roquin. It has been suggested that Regnase-1 and Roquin degrade an overlapping set of mRNAs including inflammatory mRNAs (for example, TNF-α and IL6) via distinct mechanisms. Our experiments demonstrate that TNF-α and IL6 3′ UTRs are targeted by translation-dependent degradation including endonucleolytic cleavage in HeLa cells. We mapped the cleavage sites to positions up- and downstream of the conserved stem–loop sequences (Supplementary Fig. 6). Our observations for the IL6 3′ UTR are in good agreement with recently published results, although endocleavage did not occur at the putative binding site of Regnase-1 (ref. 4). However, the prominent endonucleolytic cleavage of the TNF-α 3′ UTR was unexpected in light of the preferential Roquin-mediated deadenylation that has been reported in the literature[4,44]. Furthermore, the site of endonucleolytic cleavage was clearly downstream of the reported Roquin and Regnase-1 binding sites. Deletion of the endocleavage sites partially impairs the degradation of TNF-α and IL6 3′ UTR reporter mRNAs, but also indicates that additional pathways are involved in the turnover of the reporters. Hence, xrFrag analysis confirms that multiple decay mechanisms contribute to the efficient mRNA degradation triggered by TNF-α and IL6 3′ UTRs.

In conclusion, we have established xrFrag analysis as a powerful tool to study mRNA turnover in mammalian cells and demonstrate its functionality by applying it to several mRNA decay pathways. Therefore, we envision that the use of xrRNAs will facilitate the analysis of mRNA decay under different experimental conditions and in various biological systems.

## Methods

**Plasmids and cell culture.** Plasmid constructs β-globin WT and PTC39, TPI–WT and PTC160, TPI-4MS2 tethering reporter, LacZ control plasmid and the expression vector for MS2V5-BTZ were generated by inserting the respective DNA fragments[26,45] into the pCI-neo vector (Promega). Plasmids encoding for reporter constructs contained four copies of the northern blot probe binding site, as indicated in the figures. The MVE virus xrRNA element (nucleotides 10,488–10,725, reference sequence NC_000943.1, Fig. 1b) with a 50 nucleotide upstream linker was amplified from a gBlocks gene fragment (IDT) and inserted in the site indicated in the figures. Similarly, the 3′ UTR regions of RAB7A (nucleotides 862–1,311, reference sequence NM_004637.5), TNF-α (nucleotides 1,201–1,639, reference sequence NM_000594.3, Supplementary Fig. 6a) and IL6 (nucleotides 761–1,082, reference sequence NM_000600.4, Supplementary Fig. 6b), as well as the EMCV IRES and MALAT 3′ sequence were amplified from gBlocks gene fragments (IDT) or HeLa cDNA and cloned in the pCI vector. AREs (ref. 36) and miREs (ref. 37) were assembled from annealed primers. For stable cell lines, the reporters were cloned in the pcDNA5/FRT/TO vector. Full-length or mutants of SMG7, PNRC2, TNRC6A, TNRC6B and TTP were cloned from HeLa cDNA and inserted into the pCI-MS2V5 tethering vector. The NanoLuc ORF was cloned in pCI-neo containing a stable 5′ stem–loop (kindly provided by Gabriele Neu-Yilik), which was previously used to prevent cap-dependent translation in IRES-containing reporter constructs[46].

Transient transfections were done in HeLa Tet-Off cells (Clontech). Standard protocols were used to generate stable HeLa Flp-In T-REx cell lines (initially established by Elena Dobrikova and Matthias Gromeier, Duke University Medical Center). Expression of stable cell lines was induced with 1 µg ml⁻¹ doxycycline for 24 h or for the indicated time before harvesting. All cell lines were cultured in DMEM (Gibco) supplemented with 9% foetal bovine serum (Gibco) and 1 × Pen Strep (Gibco) and the cells were incubated at 37 °C, 5% CO2 and 90% humidity. Mycoplasma contamination was tested by PCR amplification of mycoplasma-specific genomic DNA[47].

**siRNA transfections.** For cells intended to be transiently transfected with plasmids, $5 \times 10^5$ HeLa Tet-Off cells were grown overnight in 6 cm plates and forward transfected with 300 pmol siRNA for single or 600 pmol total siRNA for double knockdowns using Lipofectamine RNAiMAX (Thermo Fisher)[26]. After 48 h the cells were counted and transferred to 6-well plates. Stable cell lines were reverse transfected using 2.5 µl Lipofectamine RNAiMAX and 60 pmol siRNA per $2 \times 10^5$ HeLa cells. The following siRNA target sequences were used: luciferase (5′-CGT ACGCGGAATACTTCGA-3′), XRN1 (5′-AGATGAACTTACCGTAGAA-3′), UPF1 (5′-GATGCAGTTCCGCTCCATT-3′), SMG6 (5′-GGGTCACAGTGC TGAAGTA-3′), SMG7 (5′-CGATTTGGAATACGCTTTA-3′), CNOT1 (5′-GGA ACUUGUUUGAAGAAUA-3′), DCP2 (5′-GGACTGGCTTTCTCGAAGA-3′), for EDC4 an equal mix of EDC4-1 (5′-GAGTTAAAGATGTGGTGTA-3′) and EDC4-2 (5′-TACACCACATCTTTAACTC-3′). The efficiency of the CNOT1 knockdown was confirmed by semi-qPCR using the following primers: 5′-AATGTTGGCCTG TCTGCAAG-3′, 5′-TGTCATTCCAGCAAGAGGGT-3′.

**Transient plasmid transfections.** $2.8 \times 10^5$ HeLa Tet-Off cells were seeded in 6-well plates 24 h before transfection by calcium phosphate precipitation with 0.5 µg of a mVenus expression plasmid, 2.5 µg control plasmid (LacZ) and 2 µg plasmid encoding for reporter mRNA. For tethering assays, 1.5 µg reporter, 1.5 µg of MS2V5-tagged expression plasmid and 3 µg control plasmid (LacZ) were transfected.

**RNA extraction and northern blotting.** Total RNA was extracted from individual replicates with peqGOLD TriFast (Peqlab), resolved on a 1% agarose and 0.4 M formaldehyde gel using the tricine-triethanolamine buffer system[48] and analysed by northern blotting. PSP65-globin plasmid was linearized with BamHI and used as template for in vitro transcribed [α-32P]-GTP body-labelled RNA probes, which were used for the detection of all reporter and LacZ control RNA. 7SL endogenous RNA was detected using a 5′-32P-labelled oligonucleotide (5′-TGCTCCGTTTCC GACCTGGGCCGGTTCACCCCTCCTT-3′). Signals were scanned using a Typhoon FLA 7000 (GE Healthcare) and raw unmodified scans were quantified using ImageQuant TL (GE Healthcare). Representative blots of at least three replicates are shown. Blots have been cropped and contrast-adjusted for presentation purposes. Uncropped images of main figures are collectively shown in Supplementary Fig. 8. As an example, positions of RNA marker bands are shown in Supplementary Figs 2 and 5. For sub-cellular fractionation, the cells were first harvested in polysome buffer (10 mM NaCl, 10 mM MgCl2, 10 mM Tris-HCl pH 7.4, 1% Triton X-100, 1% Na-deoxycholate and 1 mM DTT). Cytoplasmic and nuclear fractions were separated by centrifugation and RNA subsequently extracted from each fraction using peqGOLD TriFast. For mRNA half-life calculations, the one-phase-decay equation from GraphPad Prism 5 was used.

**Immunoblot analysis and antibodies.** SDS–polyacrylamide gel electrophoresis and immunoblot analysis were performed using protein samples derived from peqGOLD TriFast extractions or parallel transfection harvested with RIPA buffer. The antibody against tubulin (T6074; 1:3,000 dilution) was from Sigma, the antibody against V5 (18870; 1:3,000 dilution) was from QED Bioscience, the antibodies against GFP (ab290; 1:3,000 dilution) and SMG6 (ab87539; 1:1,000 dilution) were from Abcam, the antibodies against XRN1 (A300-443A; 1:2,000 dilution), SMG7 (A302-170 A; 1:2,000 dilution) and EDC4 (A300-745A; 1:1,000) were from Bethyl and the antibody against UPF1 was kindly provided by Jens Lykke-Andersen (1:1,000 dilution). Secondary horseradish peroxidase-coupled antibodies against rabbit (111-035-006; 1:3,000 dilution) or mouse (115-035-003; 1:3,000 dilution) were from Jackson ImmunoResearch. Western Lightning Plus-ECL Enhanced Chemiluminescence Substrate (PerkinElmer) in combination with the myECL Imager (ThermoFisher) was used for visualization. Blots have been cropped and contrast-adjusted for presentation purposes. Uncropped images of main figures including molecular weight markers are collectively shown in Supplementary Fig. 8.

**3′ fragment cloning.** Total RNA (25 µg) was used for linker ligation using T4 RNA ligase I and 200 pmol of barcoded RNA linker (rGrCrUrGrArUrGrGrC rGrArUrGrArArUrGrArNrNrNrNrNrNrArArA)[26]. After RT–PCR with VNN-oligo(dT)20 primer, TNF-α- or IL6-specific primers together with an adaptor-specific primer (Adaptor: 5′-GCTGATGGCGATCAATGA-3′; TNF-α: 5′-TTT TTTTGCGGCCGCCCGGGTCGACTAGTAGGGCGATTACAGACACA-3′; IL6: 5′-TTTTTTGCGGCCGCCCGGGTCGACTGAGGTAAGCCTACACTT TC-3′) were used for PCR amplification with Accuprime Taq Polymerase (Life Technologies). Amplified fragments were subsequently cloned in pGEM-T vector, sequenced by GATC Biotech and unique clones were used for mapping analysis.

**Data Availability.** The authors declare that all the data supporting the findings of this study are available within the article and its Supplementary Information files and from the corresponding authors upon reasonable request.

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

## Acknowledgements

We thank Heidi Thelen and Juliane Hancke for technical assistance; the Leptin lab for sharing equipment; Valentina Potrich for sharing protocols; Gabriele Neu-Yilik for providing the pCI-neo stem–loop vector; Elena Dobrikova and Matthias Gromeier for establishing and Matthias Hentze for sharing the HeLa Flp-In T-REx cells; Anna-Lena Steckelberg for critical reading of the manuscript and members of the Gehring lab for useful discussions. We are grateful to Jens Lykke-Andersen for the antibody against UPF1. This research was funded by a grant from the Deutsche Forschungsgemeinschaft (GE2014/4-1) to N.H.G.

## Author contributions

N.H.G and V.B. conceived the study. V.B., J.V.G., M.C.M. and N.H.G. performed the experiments. V.B., J.V.G. and N.H.G. wrote the manuscript. All authors discussed the results and edited the manuscript.

## Additional information

**Competing financial interests**: The authors declare no competing financial interests.

