## [Peer Review File · Nature Communications]

Reviewers' Comments:

Reviewer #1 (Remarks to the Author)

In this manuscript entitled "Interrogating the degradation pathways of unstable mRNAs with XRN1-resistant sequences", Boehm et al. developed a molecular tool to study 5'-3' mRNA decay via the detection of XRN1-resistant decay fragments (xrFrag) and applied it to nonsense-mediated mRNA decay (NMD) and cytokine 3' UTR degradation. The authors demonstrated that endonucleolytic cleavage within cytokine TNF and IL6 3' UTRs initiates 5'-3' decay pathway. Moreover, the authors showed that 5'-3' degradation is mainly triggered by decapping and endonucleolytic cleavage and SMG7-mediated degradation does not only enhance 5'-3' decay but also lead to the degradation of the substrate mRNA by 3'-5' decay in NMD.

Overall, this work is novel and interesting by providing an approach "xrFrag" for 5'-3' mRNA analyses. The manuscript is concisely written.

Nevertheless, there are specific comments to be addressed.

Specific comments

1. The authors applied the xrFrag approach to only NMD and cytokine 3' UTRs degradation. Because microRNA (miRNA)-mediated mRNA decay is a well-known post-transcriptional regulation of gene expression, it would be nice for the authors to apply it to miRNA-mediated mRNA decay and demonstrate whether miRNA-mediated mRNA decay enhance 5'-3' degradation.
2. In Figure 1c, 2b and 3b, reporter mRNAs without xrRNA and RNA size marker should be included as control.
3. In Figure 3, it should be shown whether tethering of PNRC2, SMG7, BTZ, TNRC6A, TNRC6B and TTP promote deadenylation of reporter mRNAs.
4. Figure 4c showed that knockdown of EDC4 and DCP2 had no effect on production of xrFrag. Because xrFrag and 3' fragments are predicted to be decreased and increased in knockdown of EDC4 and DCP2, respectively, the authors should explain why knockdown of EDC4 and DCP2 had no effect on production of xrFrag and 3' fragments.

Reviewer #2 (Remarks to the Author)

Turnover of mRNA in eukaryotic cells occurs by three distinct pathways, 5' - 3' exonucleolytic degradation, 3' - 5' exonucleolytic digestion, and endonucleolytic cleavage. In this manuscript, Niels Gehring and colleagues utilize a novel viral RNA sequence recently found to efficiently block 5' - 3' digestion by the cytoplasmic exonuclease XRN1 as a means to investigate the directionality of decay for mammalian mRNA. In particular, this work focuses on the degradation of two classes of unstable mRNAs, those harboring nonsense codons that target them to the nonsense-mediated mRNA decay (NMD) pathway, and unstable cytokine mRNAs harboring RNA stem-loop structure in their 3' UTRs that target the transcripts to endonucleolytic cleavage.

There are a number of conclusions made by the authors from this work. First, the use of RNA structural elements that block XRN1 progression provides an effective tool to monitor the directionality of mRNA decay pathways in human cells. Second, nonsense-containing mRNAs are degraded predominantly by SMG6-mediated endonucleolytic cleavage of the mRNA near the site of premature translation termination. Third, NMD factor SMG7 promotes 5' - 3' decay of nonsense-containing mRNA and this pathway is secondary to SMG6-mediated decay. Forth, in the absence of SMG6 and 5' - 3' decay activities, 3' - 5' decay is observed to also be promoted by SMG7. Fifth, the 3' UTR stemloop elements in TNFalpha and IL6 mRNAs reduce reporter mRNA levels and target the transcripts to endonucleolytic cleavage, possibly at sites distinct from those previously

reported.

While this work is the first to describe the use of XRN1-resistant sequences to analyze mRNA decay pathways in mammals, this approach is identical to that pioneered to study mRNA turnover in yeast (2 decades ago - much of this work was neither acknowledged nor referenced; using G-quadruplex structures to block XRN1 activity), and provides little new information regarding the turnover of these two particular classes of mammalian mRNA.

Specific Comments

1. The authors suggest that monitoring the accumulation of 3' RNA decay fragments resulting from XRN1 inhibition is a quantitative and robust approach to studying mRNA decay, and an acceptable alternative to measuring mRNA decay rates directly. This is inaccurate since fragment accumulation only monitors 5' - 3' degradation and it is unclear how complete this block is to digestion by XRN1 or other exonucleases (thus the accumulated product may not truly represent the contribution from 5' - 3' degradation).

2. The two main conclusions from this work regarding the decay of NMD substrates are based solely on the analysis of two reporter mRNAs and are not novel. Namely, several groups have suggested (using reporters and genome-wide approaches) that the major decay pathway for mammalian nonsense-containing mRNA is via SMG6-mediated endonucleolytic cleavage. Additionally, it has been previously shown that SMG7 promotes 5' - 3' decay of NMD substrates.

3. The conclusion that SMG7 also promotes 3' - 5' decay of NMD substrates may be misleading since it was observed only when the SMG6 and 5' - 3' pathways were inhibited. Nonetheless, 3' - 5' decay of NMD substrates has been previously reported both in mammals and in yeast.

4. It has been previously reported that TNF α and IL6 mRNAs are targeted for decay through endonucleolytic cleavage of stem-loop sequences found within their 3' UTRs. The only novel findings reported here is the possibility that cleavage is outside of the predicted sites. One should be cautious, however, to define cleavage sites based on the mapped mRNA ends, since 100% knock-down efficiencies are never achieved and it is likely that the 3' and 5' ends resulting from endonucleolytic cleavage are nibbled (in vivo and in vitro) prior to mapping.

5. It is noted:

a) Neither mRNA stability nor turnover is ever directly measured in this study and it should not be inferred from the measurement of full-length or fragment mRNA abundance.

b) Neither 3' - 5' decay nor poly(A) tail removal (deadenylation) was directly measured and thus the contribution of these events can only be suggested.

c) Reporting full length mRNA:fragment levels is misleading, particularly in cases where mRNA decay pathways have been inhibited by knock-down of particular factors.

6. Several experiments use the translational inhibitor cycloheximide to assess the contribution of translation to mRNA decay. It should be noted, however, that XRN1 activity can be blocked by the ribosome, which could contribute to the decreased accumulation of the XRN1 resistant fragment.

Reviewer #3 (Remarks to the Author)

Analysis of mRNA decay pathways is hampered by the difficulty to experimentally detect degradation intermediates. In mammalian cells, XRN1 knockdown has so far been the only way to stabilize and detect degradation intermediates undergoing 5' to 3' XRN1-mediated exonucleolysis. Boehm and colleagues show in this manuscript that the introduction of two consecutive XRN1-resistant elements from MVE (referred to as xrRNA fragments) into reporter RNAs efficiently block XRN1 and hence result in the accumulation of stable degradation intermediates. This will undoubtedly become an important tool in many future RNA turnover studies involving XRN1-mediated RNA

degradation.

While the xrRNA fragment clearly increases the amounts of the downstream RNA part, there appears to be some level of leakiness, as indicated by the presence of the xrFragB in WT constructs containing two xrRNA fragments (Fig 2h, Suppl. Fig 6b). Since the extent of this leakiness is not known, quantitative conclusion should be drawn cautiously.

The authors used xrRNA fragment-containing reporter transcripts to interrogate NMD and the accelerated degradation of IL6 and TNF-alpha mRNA. With PTC-containing NMD targets, the authors observed a strong accumulation of rRNA fragment that was dependent on translation and UPF1 - two hallmarks of NMD - and found with the beta-globin-xrRNA PTC39 construct that XRN1 depletion i) decreased the xrRNA fragment, ii) led to the detection of a new fragment consistent in size with endonucleolytic cleavage near the PTC, and iii) did not increase full length RNA abundance, indicating that the decay of most of these NMD reporter transcripts is initiated by endonucleolytic cleavage rather than decapping, consistent with previous reports from the Gehring lab and others. Interestingly, the authors also found a translation dependence of the degradation mediated by the IL6 and TNF-alpha 3' UTRs, which has not been previously appreciated. Their data further suggest that also in this case, the major degradation route starts with endonucleolytic cleavage in the respective 3' UTR. It is suspected, based in previous work, that Regnase-1 catalyzes this endonucleolytic cleavage and the authors should test this assumption by analyzing their xrRNA constructs in cells with a Regnase-1 knockdown.

The authors next combined their xrRNA fragment constructs with MS2 tethering assays. Interestingly, tethering of BTZ/MLN51 led to xrRNA fragment accumulation in a translation-dependent manner, consistent with the idea that BTZ/MLN51 promotes endonucleolytic cleavage. Tethering of PNRC2 led to accumulation of xrRNA fragment and full length RNA in a translation-independent manner, supporting the view that PNRC2 promotes decapping of the target RNA. Finally, tethering of the C-terminal part of SMG7, which was previously shown to interact with the CCR4/NOT deadenylase complex, led to a translation-independent decrease of both the full length transcript and the xrRNA fragment, indicating that a fraction of this decay might occur by 3'-to-5' exonucleolysis. Supporting evidence for this interpretation came from a construct harboring the MALAT triple-helix 3' end, which confers resistance to 3'-to-5' exonucleolysis (including deadenylation). Similar results were obtained by tethering of other deadenylation promoting factors like TNRC6A/B and TTP, letting the authors to conclude that a substantial amount of degradation of deadenylated RNA occurs by 3'-to-5' pathways.

In the final part of the manuscript, the authors used their xrRNA reporters to address the relative contributions of SMG6- and SMG7-mediated degradation pathways to NMD. Confirming earlier reports, their results showed that for all three tested NMD reporters, SMG6 was the major and SMG7 the minor degradation-promoting factor. However, I don't understand the interpretation of the result shown in Suppl. Fig. 6b (described in lines 254-258). To me it looks as if the WT construct is mainly degraded by decapping followed by XRN1-mediated 5'-to-3' exonucleolysis, which leads to the accumulation of xrFragA and, because of certain leakiness of the XRN1 block, to some xrFragB. By contrast, essentially all decay seems triggered by endonucleolysis near the PTC in the PTC160 construct, leading to a strong accumulation of xrFragB and a disappearance of xrFragA, which does not prevent 3'-to-5' exonucleolytic decay of the 5' decay intermediate. Is there anything that speaks against this much simpler interpretation of this result?

Overall, the results shown are of high quality and the conclusions are compelling (with exception of the points mentioned above). I found the manuscript a bit difficult to read because of the jumping back and forth between NMD and other decay pathways. I urge the authors to test whether another grouping of the data would possibly allow for a more streamlined story telling. The highlight of this work is clearly the presentation of the xrRNA fragment system as an elegant tool that allows stabilization of RNA degradation intermediates, whereas the results do not provide any fundamentally new insights into the interrogated decay pathways but are mostly confirming

previous findings.

Minor points:

Fig 1b: what is the difference between constructs Full and 1+2? Please specify.

Suppl. Fig 1: The greek symbol alpha in TNF-alpha is not displayed correctly in the figure legend (Presumably a conversion problem between MS Word and pdf.

Suppl. Fig. 2F: TPI should be labeled TPI-xrRNA to be consistent with the other panels.

p. 9, lines 189-192 and Fig 3c: It looks as if the decreased xrRNA fragment amounts with tethered SMG7-FL or SMG7-C are barely significant. To conclude from this marginal decrease that this is in opposition to the current model of deadenylation-dependent decapping seems a far stretch.

Reviewers' comments:

Reviewer #1 (Remarks to the Author):

In this manuscript entitled "Interrogating the degradation pathways of unstable mRNAs with XRN1-resistant sequences", Boehm et al. developed a molecular tool to study 5'-3' mRNA decay via the detection of XRN1-resistant decay fragments (xrFrag) and applied it to nonsense-mediated mRNA decay (NMD) and cytokine 3' UTR degradation. The authors demonstrated that endonucleolytic cleavage within cytokine TNF and IL6 3' UTRs initiates 5'-3' decay pathway. Moreover, the authors showed that 5'-3' degradation is mainly triggered by decapping and endonucleolytic cleavage and SMG7-mediated degradation does not only enhance 5'-3' decay but also lead to the degradation of the substrate mRNA by 3'-5' decay in NMD.

Overall, this work is novel and interesting by providing an approach "xrFrag" for 5'-3' mRNA analyses. The manuscript is concisely written.

- We thank the reviewer for this positive evaluation of our work.

Nevertheless, there are specific comments to be addressed.

Specific comments

1. The authors applied the xrFrag approach to only NMD and cytokine 3' UTRs degradation. Because microRNA (miRNA)-mediated mRNA decay is a well-known post-transcriptional regulation of gene expression, it would be nice for the authors to apply it to miRNA-mediated mRNA decay and demonstrate whether miRNA-mediated mRNA decay enhance 5'-3' degradation.

- We agree with reviewer #1 that miRNA-mediated mRNA decay represents an important regulatory process and therefore is an interesting candidate for our xrFrag analysis. To this end, we constructed new reporters containing dual miRNA responsive elements (miRE) targeted by highly expressed miRNAs in HeLa cells (let-7 and miR-21). To control for the specificity of miRNA-mediated decay, we compared WT binding sites to mutated binding sites. We placed these elements between the TPI ORF and the xrRNA (Fig. 5e) and generated HeLa stable cell lines expressing these reporters. Although the miRE-WT reporter exhibited strongly reduced steady state reporter levels, we could not detect a substantial increase in xrFrag levels, compared to the miRE-Mut reporter (Fig. 5f). This indicated that the miRNA-induced degradation occurs primarily via deadenylation and/or 3'-5' decay. We furthermore show that the knockdown of CNOT1 (scaffolding component of the CCR4-NOT complex) stabilizes the reporter, thereby underlining our observation (Fig. 5g). In conclusion, our xrFrag analyses supports the current model, in which miRNA-mediated decay is initiated by deadenylation. Our results further suggest that the subsequent degradation of the mRNA occurs via 5'-3' as well as 3'-5' decay.
- To further explore the applicability of the xrFrag method, we also constructed stable cell lines expressing reporter mRNAs containing individual AU-rich elements (AREs). For this purpose, we chose the c-fos and GM-CSF AREs as suitable sequence elements (Fig. 5a). Interestingly, and in contrast to miRNA-mediated decay, we observed increased xrFrag upon degradation of the ARE-containing transcripts (Fig. 5b). Of note, the decay of the mRNA and the accumulation of xrFrag of TPI-GM-CSF was dependent on deadenylation (Fig. 5c), pointing to a deadenylation-dependent enhancement of 5'-3' degradation.
- In combination, we thank the reviewer for the stimulating comment, which lead us to identify two deadenylation-dependent degradation pathways, which culminate in a differential induction of subsequent decay routes.

2. In Figure 1c, 2b and 3b, reporter mRNAs without xrRNA and RNA size marker should be included as control.

- We presume that the motivation behind this comment is (a) to confirm the identity of the bands on the northern blots and (b) to exclude non-specific effects of the insertion of the xrRNA sequence. In fact, we are able to unequivocally identify all bands, because there are no non-specific bands visible on the northern blots, which are all shown in the Figures. Furthermore, the insertion of the xrRNA sequence did not alter the behavior of any reporter mRNA we have investigated (for example Fig. 1e, 2b). Nonetheless, we included two novel figures with RNA size markers as control (Fig. 1e and Supplementary Fig. 4e) and have also repeated several old experiments and could confirm that all our indicated RNAs fit to the estimated sizes. However, we believe that including size markers and reporters +/- xrRNA in most of our experiments would unnecessarily expand the manuscript and make the figures more difficult to understand.

3. In Figure 3, it should be shown whether tethering of PNRC2, SMG7, BTZ, TNRC6A, TNRC6B and TTP promote deadenylation of reporter mRNAs.

- Reviewer #1 raises an important point, which we addressed by time-course and knockdown assays. However, we experienced technical difficulties with these experiments, as CNOT1 knockdown cells could not be transfected efficiently and we could not chase the deadenylation of the reporter mRNAs in transcriptional shutoff experiments. Nevertheless, we had observed in the beginning of this project, that swapping the xrRNA and 4MS2 elements in a reporter (Supplementary Fig. 4d) resulted in an observable shift of the xrFrag, depending on the tethered protein. We speculate that in this arrangement of RNA elements the xrFrag will be more prone to deadenylation by tethering of deadenylating proteins, because the 4MS2 sites are retained in the xrFrag. This effect might be lost in the reporter with the 4MS2-xrRNA cassette, which was used for all other tetherings. To test this, we repeated the experiments with potential deadenylation-promoting factors using the xrRNA-4MS2 reporter. Indeed, we observed a clear shift of the xrFrag when SMG7-C or the silencing domains of TNRC6A or B were tethered to the transcript (Supplementary Fig. 4e,f). We conclude that maintaining the interaction of these proteins with the decay intermediate results in constant activation of deadenylation, leading to the observable size difference in northern blots.

4. Figure 4c showed that knockdown of EDC4 and DCP2 had no effect on production of xrFrag. Because xrFrag and 3' fragments are predicted to be decreased and increased in knockdown of EDC4 and DCP2, respectively, the authors should explain why knockdown of EDC4 and DCP2 had no effect on production of xrFrag and 3' fragments.

- We do not expect to see an increase of 3' fragment production upon EDC4 and DCP2 (Figure 3c, lane 6), which would only be visible in combination with XRN1 knockdown. However, the functional effect of the EDC4 and DCP2 knockdown is detectable in combination with the SMG6 knockdown, where we observe a decrease in xrFrag/reporter ratio from 5.1 to 3.1 (Figure 3c, lanes 10&12). The lack of an effect of the EDC4 and DCP2 knockdown is like due to the dominant degradation of the reporter mRNA by endonucleolytic cleavage.

Reviewer #2 (Remarks to the Author):

Turnover of mRNA in eukaryotic cells occurs by three distinct pathways, 5' - 3' exonucleolytic degradation, 3' - 5' exonucleolytic digestion, and endonucleolytic cleavage. In this manuscript, Niels

Gehring and colleagues utilize a novel viral RNA sequence recently found to efficiently block 5' - 3' digestion by the cytoplasmic exonuclease XRN1 as a means to investigate the directionality of decay for mammalian mRNA. In particular, this work focuses on the degradation of two classes of unstable mRNAs, those harboring nonsense codons that target them to the nonsense-mediated mRNA decay (NMD) pathway, and unstable cytokine mRNAs harboring RNA stem-loop structure in their 3' UTRs that target the transcripts to endonucleolytic cleavage.

There are a number of conclusions made by the authors from this work. First, the use of RNA structural elements that block XRN1 progression provides an effective tool to monitor the directionality of mRNA decay pathways in human cells. Second, nonsense-containing mRNAs are degraded predominantly by SMG6-mediated endonucleolytic cleavage of the mRNA near the site of premature translation termination. Third, NMD factor SMG7 promotes 5' - 3' decay of nonsense-containing mRNA and this pathway is secondary to SMG6-mediated decay. Forth, in the absence of SMG6 and 5' - 3' decay activities, 3' - 5' decay is observed to also be promoted by SMG7. Fifth, the 3' UTR stemloop elements in TNFalpha and IL6 mRNAs reduce reporter mRNA levels and target the transcripts to endonucleolytic cleavage, possibly at sites distinct from those previously reported.

While this work is the first to describe the use of XRN1-resistant sequences to analyze mRNA decay pathways in mammals, this approach is identical to that pioneered to study mRNA turnover in yeast (2 decades ago - much of this work was neither acknowledged nor referenced; using G-quadruplex structures to block XRN1 activity), and provides little new information regarding the turnover of these two particular classes of mammalian mRNA.

- We are aware of the previous use of G-quadruplex sequences to investigate 5'-3' exonucleolytic decay in yeast and highly appreciate the pioneering work of the Parker and Raue labs. However, to the best of our knowledge, neither G-quadruplex sequences nor other XRN1-inhibitory sequences have been successfully used in mammalian cells before. Admittedly, we did not reference work using G quadruplexes in the initial version of the manuscript. The reason for this was neither willful ignorance nor neglect, but simply the fact that mRNA turnover seems to be very different in yeast and humans (Lim et al., Cell. 2014 Dec 4;159(6):1365-76. doi: 10.1016/j.cell.2014.10.055.). As a proof on concept for Xrn1p inhibition in yeast we include appropriate references in the revised version of the manuscript (l. 324-327).
- Our xrRNA analysis is not restricted to NMD substrates or cytokine mRNAs, but can be used to study different types of mammalian mRNA turnover. We are therefore convinced that our work represents an important technical improvement to the existing methodology and will enable to dissect mRNA decay pathways of all classes of unstable mRNAs.

Specific Comments

1. The authors suggest that monitoring the accumulation of 3' RNA decay fragments resulting from XRN1 inhibition is a quantitative and robust approach to studying mRNA decay, and an acceptable alternative to measuring mRNA decay rates directly. This is inaccurate since fragment accumulation only monitors 5' - 3' degradation and it is unclear how complete this block is to digestion by XRN1 or other exonucleases (thus the accumulated product may not truly represent the contribution from 5' - 3' degradation).

- Although we appreciate the comment from reviewer #2, we do not completely agree with it. According to the current view of eukaryotic mRNA degradation, decapping, endocleavage as well as deadenylation (in part) are expected to lead to enhanced 5'-3' decay. Since our xrFrag analysis monitors 5'-3' degradation by blocking XRN1 it provides a semi-quantitative measure

of mRNA turnover and is therefore superior to methods that only measure the relative levels of full length transcripts. In contrast to measuring decay rates (i.e. half-life times), the xrRNA analysis also determines the directionality of the decay. Hence, xrRNA analysis is an ideal starting method if only little information about the decay pathway of a given mRNA is available. We hope to persuade the reviewer that the xrRNA analysis has several advantages over traditional methods by analyzing in total four different degradation pathways (NMD, cytokine-related decay, miRNA-mediated decay and ARE-mediated decay) with this method. Our data indicate that under certain conditions xrRNA analysis can be used to monitor deadenylation, which would enable to study 3'-5' decay.

2. The two main conclusions from this work regarding the decay of NMD substrates are based solely on the analysis of two reporter mRNAs and are not novel. Namely, several groups have suggested (using reporters and genome-wide approaches) that the major decay pathway for mammalian nonsense-containing mRNA is via SMG6-mediated endonucleolytic cleavage. Additionally, it has been previously shown that SMG7 promotes 5' - 3' decay of NMD substrates.

- This comment of reviewer #2 requires some clarification. Indeed, it has been shown previously that NMD is mainly initiated by endonucleolytic cleavage. This fact was explicitly mentioned in the original version of the manuscript (lines 142-143 "Recent work demonstrated that SMG6-catalyzed endocleavage represents the major degradation pathway of NMD substrates in mammalian cells²⁴"). We consider our analysis of endonucleolytic cleavage during NMD as a control for the xrRNA method. The same is true for 5'-3' decay of NMD substrates. However, we observe some discrepancy with the previously proposed notion that accelerated deadenylation of NMD substrates exclusively enhances subsequent decapping and 5'-3' degradation.
- It is inherent in the xrRNA analysis that it is currently only used in low to medium throughput assays. However, we have generated more NMD reporter constructs (one is shown for example in Fig. 2g-h), which display the expected behavior.

3. The conclusion that SMG7 also promotes 3' - 5' decay of NMD substrates may be misleading since it was observed only when the SMG6 and 5' - 3' pathways were inhibited. Nonetheless, 3' - 5' decay of NMD substrates has been previously reported both in mammals and in yeast.

- The remarkable efficiency of NMD is one of its fascinating characteristics. Even in the absence of an important NMD factor like SMG6 only a partial accumulation of NMD substrates is observed. It seems therefore legitimate to analyze NMD under SMG6 depletion conditions, in order to disentangle the different degradation pathways. This approach is also commonly used by other labs in the NMD field.
- It should also be noted that some organisms lack certain genes of the NMD pathway, for example drosophila does not contain SMG6 (Gatfield et al., EMBO J. 2003 Aug 1;22(15):3960-70.), whereas some plants seem to lack SMG5 and SMG6 (Krenyi et al., EMBO J. 2008 Jun 4;27(11):1585-95. doi: 10.1038/emboj.2008.88 ; Shaul, Trends Plant Sci. 2015 Nov;20(11):767-79. doi: 10.1016/j.tplants.2015.08.011..). Furthermore, inhibitors of NMD that are targeting SMG7 have been described (Martin et al, Cancer Res. 2014 Jun 1;74(11):3104-13. doi: 10.1158/0008-5472.CAN-13-2235.). Overall, it appears to be important to understand the minor decay activity of SMG7 in order to understand NMD.
- Admittedly, 3'-5' decay has been previously reported both in mammals and in yeast. However, no specific NMD factor has been assigned to this decay activity.

4. It have been previously reported that TNFalpha and IL6 mRNAs are targeted for decay though endonucleolytic cleavage of stem-loop sequences found within their 3' UTRs. The only novel findings

reported here is the possibility that cleavage is outside of the predicted sites. One should be cautious, however, to define cleavage sites based on the mapped mRNA ends, since 100% knock-down efficiencies are never achieved and it is likely that the 3' and 5' ends resulting from endonucleolytic cleavage are nibbled (in vivo and in vitro) prior to mapping.

- As described in the manuscript, we deliberately chose the TNF- α and IL6 3' UTRs, because their turnover has been characterized before. We were surprised to obtain new insights into the degradation of these two mRNAs, albeit they have been intensely studied for many years. In our view this demonstrates the analytical strength of the xrRNA method.
- After the analysis of new deletion mutants of TNF- α and IL6 we are now confident that we have identified the bona fide cleavage sites in both 3' UTRs. In brief, we found that TNF- α and IL6 constructs that contain the conserved stem loop, but lack the upstream/downstream cleavage sites, show a reduction of xrFrag levels compared to the full length 3' UTR construct (Fig. 7e). Furthermore, endocleavage within the TNF- α 3' UTR occurs even in the absence of the stem loop (Fig. 7f), i.e. only requires the sequences in the vicinity of the cleavage site. We therefore propose that TNF- α endocleavage must occur via an alternative pathway that does not act via the putative Reg-1 binding site. We consider this analysis a convincing example for the application of xrRNA.
- Since endocleavage within the TNF- α 3'UTR is independent of the stem loop, we would exclude extensive nibbling and mis-identification of the site of endocleavage.

5. It is noted: a) Neither mRNA stability nor turnover is never directly measured in this study and it should not be inferred from the measurement of full-length or fragment mRNA abundance. b) Neither 3' - 5' decay nor poly(A) tail removal (deadenylation) was directly measured and thus the contribution of these events can only be suggested. c) Reporting full length mRNA:fragment levels is misleading, particularly in cases where mRNA decay pathways have been inhibited by knock-down of particular factors.

- We would like to thank reviewer #2 for this comment, which has motivated us to directly measure the turnover of three different reporter mRNAs (Fig. 2h). This experiment confirms that PTC-containing NMD substrates are rapidly degraded (which was known before), whereas the resulting xrFrag decays with a much slower half life time. An NMD substrate with a long 3' UTR, however, has a longer half-life time than a PTC-containing substrate and decays with a seemingly different kinetic. Again, the resulting xrFrag displays a slower turnover. These results are in line with the previously published literature and demonstrate the proper function of our assays.
- Furthermore, we now include a different tethering reporter (TPI-xrRNA-4MS2; Supplementary Fig. 4d), which allowed us to detect the deadenylation activity of SMG7, TNRC6A and TNRC6B (Supplementary Fig. 4e,f).

6. Several experiments use the translational inhibitor cycloheximide to assess the contribution of translation to mRNA decay. It should be noted, however, that XNR1 activity can be blocked by the ribosome, which could contribute to the decreased accumulation of the XNR1 resistant fragment.

- It should be noted that we did not detect additional "xrFrag" bands when cells were treated with CHX. Such additional bands would be expected, because the 5' UTR of the reporter mRNA is not protected by CHX-stalled ribosomes. To further prove that our observations were correct, we performed additional experiments. For TNF- α and IL6 3' UTR reporter mRNAs, we repeated our xrRNA analysis using the translation inhibitor puromycin instead of cycloheximide. Compared to cycloheximide, inhibition of translation by puromycin acts by a completely different mechanism, which results in translation termination and loss of

polysomes. Because we observed almost identical results with cycloheximide and puromycin (compare Fig. 3e and f), we conclude that the reduction in xrFrag is indeed a result of translation dependency and is not a technical artifact.

- NMD was shown numerous times in the literature to be a translation-dependent process, which we did not intend to confirm once more. We have also performed several specificity controls for PTC-containing mRNAs besides cycloheximide such as various knockdowns of NMD-factors. During the revision we carried out experiments using caffeine, a well characterized inhibitor of the NMD kinase SMG1. We found a dose-dependent inhibition of reporter degradation along with decreased xrFrag abundance (Supplementary Fig. 2g), further proving the specificity of our assays.

Reviewer #3 (Remarks to the Author):

Analysis of mRNA decay pathways is hampered by the difficulty to experimentally detect degradation intermediates. In mammalian cells, XRN1 knockdown has so far been the only way to stabilize and detect degradation intermediates undergoing 5' to 3' XRN1-mediated exonucleolysis. Boehm and colleagues show in this manuscript that the introduction of two consecutive XRN1-resistant elements from MVE (referred to as xrRNA fragments) into reporter RNAs efficiently block XRN1 and hence result in the accumulation of stable degradation intermediates. This will undoubtedly become an important tool in many future RNA turnover studies involving XRN1-mediated RNA degradation.

- We thank reviewer #3 for this evaluation and we are confident that the xrRNA analysis paves the way for future in depth analysis of mRNA decay pathways.

While the xrRNA fragment clearly increases the amounts of the downstream RNA part, there appears to be some level of leakiness, as indicated by the presence of the xrFragB in WT constructs containing two xrRNA fragments (Fig 2h, Suppl. Fig 6b). Since the extent of this leakiness is not known, quantitative conclusion should be drawn cautiously.

- This was an important comment of reviewer #3, which motivated us to estimate the efficiency of the xrRNA-mediated inhibition of XRN1. To this end, we used reporters containing multiple repetitions of xrRNA elements (Fig. 1d). Surprisingly, we found that a single xrRNA element is the most efficient inhibitor of XRN1, ergo results in the highest quantity of xrFrag (Fig. 1e). Combining two or three xrRNA elements does not generate more total xrFrag, but rather leads to the production of additional xrFrag with reduced intensity (Fig. 1e, f). This indicates that multiple xrRNAs do not cooperate to block XRN1. Nevertheless, we used the multiple xrFrag to estimate the leakiness of one xrRNA element, which is about 60 %. We are therefore aware that our xrRNA elements do not offer a complete protection from XRN1 and absolute quantitative measurements are potentially misleading. This leakiness may indeed be beneficial for the analysis, because it avoids stalling of large quantities of XRN1 at xrFrag, which would lead to the depletion of cellular XRN1. Furthermore, we are convinced that the quantification of xrFrag enables to deduct qualitative statements about mRNA degradation pathways.

The authors used xrRNA fragment-containing reporter transcripts to interrogate NMD and the accelerated degradation of IL6 and TNF-alpha mRNA. With PTC-containing NMD targets, the authors observed a strong accumulation of rRNA fragment that was dependent on translation and UPF1 - two hallmarks of NMD - and found with the beta-globin-xrRNA PTC39 construct that XRN1 depletion i) decreased the xrRNA fragment, ii) led to the detection of a new fragment consistent in size with endonucleolytic cleavage near the PTC, and iii) did not increase full length RNA abundance, indicating that the decay of most of these NMD reporter transcripts is initiated by endonucleolytic cleavage

rather than decapping, consistent with previous reports from the Gehring lab and others. Interestingly, the authors also found a translation dependence of the degradation mediated by the IL6 and TNF-alpha 3' UTRs, which has not been previously appreciated. Their data further suggest that also in this case, the major degradation route starts with endonucleolytic cleavage in the respective 3' UTR. It is suspected, based in previous work, that Regnase-1 catalyzes this endonucleolytic cleavage and the authors should test this assumption by analyzing their xrRNA constructs in cells with a Regnase-1 knockdown.

- We agree with reviewer #3 that the participation of Regnase-1 (Reg-1) in the observed endonucleolytic degradation pathway should be analyzed. To address this point, we performed siRNA-mediated knockdown of Reg-1 in combination with XRN1 knockdown in cell lines expressing the cytokine 3' UTR reporters. We used the Reg-1 siRNA targeting sequence reported previously (Mino et al., Cell 2015, Volume 161, Issue 5, Pages 1058–1073). Unfortunately, we observed no difference in the 3' endocleavage fragment levels between the XRN1 and XRN1/Reg-1 knockdowns. Next, we repeated the experiment using a set of four pooled siRNAs targeting Reg-1 (obtained from Qiagen: FlexiTube GeneSolution GS80149 for ZC3H12A). Again, we did not observe a major decrease of the 3' endocleavage fragment (see Figure below). Analyzing the Reg-1 protein and mRNA levels by Western blotting and semi-quantitative PCR showed that neither protein nor mRNA levels were decreased following siRNA-mediated knockdown of Reg-1.

(a) HeLa FT T-REx cells expressing the indicated constructs were harvested, and the RNA was extracted and analyzed by Northern blotting. The cells were transfected with the indicated siRNAs 72 h before induction. Expression of the reporter constructs was induced with 1 μ g/ml doxycycline. The protein levels of Reg-1 and β -actin (control) were detected by Western blotting. Mean values of reporter and xrFrag signal \pm SD ($n = 3$) were quantified and for each reporter the values of the combined XRN1/Reg-1 were normalized to the XRN1 knockdown. The ratio of xrFrag to reporter mRNA levels is indicated below the graph. (b) The mRNA levels of the of Reg-1 and GAPDH (control) were assessed by semi-quantitative PCR following reverse transcription.

- Currently, we do not entirely understand why we were not able to deplete Reg-1 in our HeLa cells by siRNA transfection, which would have been required to directly show the involvement of Reg-1 in the degradation of the cytokine 3' UTR transcripts. Our mapping of cleavage sites showed, however, that TNF- α and IL6 are cleaved up- or downstream of their conserved stem loop structure, which is the putative binding site of Reg-1. We generated deletion mutants of the TNF- α and IL6 3' UTRs in order to assess the role of the stem loop structure in the degradation of these transcripts (Fig. 7d). Surprisingly, TNF- α and IL6 constructs that contain the stem loop, but lack the upstream/downstream cleavage sites, show a reduction of xrFrag levels compared to the full length 3' UTR construct (Fig. 7e).

Furthermore, endocleavage within the TNF- α 3' UTR is still detectable in the absence of the stem loop (Fig. 7f). We therefore propose that TNF- α endocleavage must occur via an alternative pathway that does not act via the putative Reg-1 binding site. We could not perform the same experiment with the IL6 stem loop, because its cleavage site is much closer to the stem loop than that in the TNF- α 3' UTR.

The authors next combined their xrRNA fragment constructs with MS2 tethering assays. Interestingly, tethering of BTZ/MLN51 led to xrRNA fragment accumulation in a translation-dependent manner, consistent with the idea that BTZ/MLN51 promotes endonucleolytic cleavage. Tethering of PNRC2 led to accumulation of xrRNA fragment and full length RNA in a translation-independent manner, supporting the view that PNRC2 promotes decapping of the target RNA. Finally, tethering of the C-terminal part of SMG7, which was previously shown to interact with the CCR4/NOT deadenylase complex, led to a translation-independent decrease of both the full length transcript and the xrRNA fragment, indicating that a fraction of this decay might occur by 3'-to-5' exonucleolysis. Supporting evidence for this interpretation came from a construct harboring the MALAT triple-helix 3' end, which confers resistance to 3'-to-5' exonucleolysis (including deadenylation). Similar results were obtained by tethering of other deadenylation promoting factors like TNRC6A/B and TTP, letting the authors to conclude that a substantial amount of degradation of deadenylated RNA occurs by 3'-to-5' pathways.

- As discussed in detail above (Reviewer #1), we now include also a different tethering reporter (TPI-xrRNA-4MS2; Supplementary Fig. 4d), which allowed us to detect the deadenylation activity of three different proteins (Supplementary Fig. 4e,f).

In the final part of the manuscript, the authors used their xrRNA reporters to address the relative contributions of SMG6- and SMG7-mediated degradation pathways to NMD. Confirming earlier reports, their results showed that for all three tested NMD reporters, SMG6 was the major and SMG7 the minor degradation-promoting factor. However, I don't understand the interpretation of the result shown in Suppl. Fig. 6b (described in lines 254-258). To me it looks as if the WT construct is mainly degraded by decapping followed by XRN1-mediated 5'-to-3' exonucleolysis, which leads to the accumulation of xrFragA and, because of certain leakiness of the XRN1 block, to some xrFragB. By contrast, essentially all decay seems triggered by endonucleolysis near the PTC in the PTC160 construct, leading to a strong accumulation of xrFragB and a disappearance of xrFragA, which does not prevent 3'-to-5' exonucleolytic decay of the 5' decay intermediate. Is there anything that speaks against this much simpler interpretation of this result?

- We thank the reviewer for this alternative interpretation, which we find conceivable and discuss in the revised version of the manuscript (l. 168-169). Nonetheless, we would expect to observe increased amounts of xrFragA for PTC160 (compared to WT) in the absence of SMG6 (Supplementary Figure 4c, compare lanes 3&4). In our view, a likely explanation for the low amounts of xrFragA in lane 4 (accompanying low levels of the reporter mRNA) is an enhanced 3'-5' decay of the xrFragA due to its translatability.

Overall, the results shown are of high quality and the conclusions are compelling (with exception of the points mentioned above). I found the manuscript a bit difficult to read because of the jumping back and forth between NMD and other decay pathways. I urge the authors to test whether another grouping of the data would possibly allow for a more streamlined story telling. The highlight of this work is clearly the presentation of the xrRNA fragment system as an elegant tool that allows stabilization of RNA degradation intermediates, whereas the results do not provide any fundamentally new insights into the interrogated decay pathways but are mostly confirming previous findings.

- We appreciate the suggestion of the reviewer. Our original idea was to confirm the robustness of the xrRNA analysis with different unstable mRNAs in the first part of the manuscript and continue with the in-depth analysis of NMD in the second part of the manuscript. In light of the large number new results we have now rearranged the manuscript and present the different decay pathways separately, starting with our study of NMD, followed by the analysis of miRNAs, AREs and cytokine 3' UTRs.

Minor points:

Fig 1b: what is the difference between constructs Full and 1+2? Please specify.

- The difference between these two constructs is a short spacer (indicated in Supplementary Fig. 1a) located upstream of the xrRNA 1. We have now added this information to the revised manuscript (Figure legend of Fig.1b).

Suppl. Fig 1: The greek symbol alpha in TNF-alpha is not displayed correctly in the figure legend (Presumably a conversion problem between MS Word and pdf).

- The error has indeed escaped our notice. We thank the reviewer for this important information and will ensure the correct conversion of the revised manuscript.

Suppl. Fig. 2F: TPI should be labeled TPI-xrRNA to be consistent with the other panels.

- This mistake has been corrected.

p. 9, lines 189-192 and Fig 3c: It looks as if the decreased xrRNA fragment amounts with tethered SMG7-FL or SMG7-C are barely significant. To conclude from this marginal decrease that this is in opposition to the current model of deadenylation-dependent decapping seems a far stretch.

- We believe that the amounts of xrFrag from tethered SMG7 do not support the current model of deadenylation-dependent decapping. In fact, our data are consistent with the hypothesis that after the initial deadenylation by SMG7, decapping followed by 5'-3' decay as well as 3'-5' decay independent of decapping are possible degradation pathways. We are aiming at a balanced interpretation of the results in the revised version of the manuscript and therefore rephrased our conclusions.

Reviewers' Comments:

Reviewer #1 (Remarks to the Author)

The revised manuscript is substantially improved. I believe that this manuscript is ready for publication.

Reviewer #2 (Remarks to the Author)

Interrogating the degradation pathways of unstable mRNAs with XRN1-resistant sequences
Boehm et al.

This revised manuscript by Boehm et al. attempts to address several of the concerns raised in the initial submission.

- Additional experimental data using reporter mRNAs harboring miRNA binding sites and AU-rich elements provide further evidence of the broad applicability of using xrRNA elements as a tool to study the decay of unstable RNAs in mammals.
- Deletion mutants of TNF-alpha and IL6 3' UTRs provide additional evidence that the cleavage sites are bona fide sites of RNA cleavage.
- The use of puromycin as translational inhibitor confirms the observation reported in the initial submission that decreased accumulation of the xrFrag upon treatment with cycloheximide does not result from stalled ribosomes blocking XRN1.

While these revisions serve to demonstrate the utility of XRN1-blocking RNA elements in the study of mRNA turnover, they fail to address the major concerns associated with this study. Specifically:

1. The technique/approach fails to lead to any significant insight or advance in our current understanding of RNA decay in mammals. The authors purport that two 'novel' observations were gained from their study. The first - the contribution of SMG7 to 3'-5' decay to NMD - was observed only when all alternative (major) decay activities were depleted from the cell, and likely represents a by-pass pathway that only occurs under these non-physiological conditions. No evidence is provided to suggest that this pathway contributes to the turnover of NMD substrates under normal conditions. The second - the characterization of novel cleavage sites in cytokine 3' UTRs - led the authors to suggest that cleavage was not mediated by Regnase-1 and that an alternative mechanism is in play for the decay of TNFalpha mRNA. While this initial observation is intriguing, it is premature (based on the limited evidence) and the authors should be cautioned to question published findings. Indeed, the burden of proof lies with these authors to fully characterize these fragments and the mechanism for the decay of this mRNA before jumping to conclusions.

2. It needs to be re-iterated that quantitative conclusions cannot be drawn from monitoring ratios of xrFrag to full-length RNA abundance. (Notably, his measurement is also not 'semi-quantitative', as suggested by the authors). This is particularly a concern considering new data indicating that there is a considerable amount of 'leakiness' in the ability of xrRNA elements to block XRN1 and that the extent of the block to XRN1 by these elements likely differs from construct to construct (as mRNA sequence elements will likely influence the formation and/or stability of the xrRNA element and its ability to impede XRN1 progression). This reviewer is in complete disagreement that xrFrag abundance offers a superior approach for studying mRNA turnover over the gold standard of measuring mRNA stability of full-length mRNA. It does, however, provide a complementary approach that should be of interest to those in the field.

Other comments

- It is unclear why, in several figures, the abundance of the xrFrag is not significantly reduced after siRNA knockdown of XRN1 (e.g. Figs. 2b, 3c, 5c, Supp Figs. 2b, 3e). This would seem to indicate that the xrRNA element blocks other exonucleases in addition to XRN1.
- Measurement of xrFrag half-lives upon treatment with ActD is deceiving since steady-state fragment abundance is a consequence of its production from the turnover of full-length mRNA and the kinetics of degradation of the fragment itself. Notably, xrFrag production is not inhibited until

its precursor, the full-length RNA, is sufficiently depleted in the cell. Thus, any half-life value calculated from this approach represents an overestimate of the stability of the RNA fragment and of little value.

- One criticism of the initial submission was that deadenylation and 3'-5' decay are not directly measured and their contributions can only be inferred by this method. Despite the fact that the authors introduce new data that is provided to address this concern, neither mRNA polyA tail lengths nor mRNA 3' ends are directly evaluated and, thus, observations related to these processes are indirect and conclusions still only inferred.
- It is unclear why the addition of one or two additional xrRNA elements actually appears to decrease the efficiency of the most 5' element to block XRN1 activity (note the reduction in xrFragA levels in Fig 1e). This may indicate that the additional elements either lead to decay of the reporter mRNA to alternative (non-5'-3') pathways or that these elements can alter the formation/stability of the most 5' element (as suggested above).

Point-by-point response 2nd revision NCOMMS-16-04462-T

Please find below our detailed response to the comments by reviewer #2. Since there are many
changes in the text, we would like to briefly highlight the most important improvements:

- 1. Analysis of several new reporter constructs demonstrates that sequences upstream of the
xrRNA do not influence xrFrag accumulation (page 7).
- 2. Sub-cellular fractionation indicates that during NMD xrFrag are mainly produced in the
cytoplasm (page 8).
- 3. Half-life measurements establish that 3' fragments are converted into xrFrag. This explains
why xrFrag are detectable when XRN1 is depleted (page 9).

Reviewer #2 (Remarks to the Author):

Interrogating the degradation pathways of unstable mRNAs with XRN1-resistant sequences

Boehm et al.

This revised manuscript by Boehm et al. attempts to address several of the concerns raised in the
initial submission.

• Additional experimental data using reporter mRNAs harboring miRNA binding sites and AU-rich
elements provide further evidence of the broad applicability of using xrRNA elements as a tool to
study the decay of unstable RNAs in mammals.

• Deletion mutants of TNF-alpha and IL6 3' UTRs provide additional evidence that the cleavage sites
are bona fide sites of RNA cleavage.

• The use of puromycin as translational inhibitor confirms the observation reported in the initial
submission that decreased accumulation of the xrFrag upon treatment with cycloheximide does not
result from stalled ribosomes blocking XRN1.

While these revisions serve to demonstrate the utility of XRN1-blocking RNA elements in the study of
mRNA turnover, they fail to address the major concerns associated with this study. Specifically:

1. The technique/approach fails to lead to any significant insight or advance in our current
understanding of RNA decay in mammals.

• We would like to thank the reviewer for the statement that our previous revisions
“demonstrate the utility of XRN1-blocking RNA elements in the study of mRNA turnover”
(see above). This evaluation of reviewer #2 is in line with the editorial recommendation to
concentrate the manuscript on methodological advances and the application of xrFrag
analysis.

• The revised version of the manuscript specifically focuses on the broad applicability of the
xrFrag method. This has been achieved by analyzing multiple important mammalian decay
pathways, which due to space restrictions precludes in-depths analyses of every individual
pathway. Nonetheless, the manuscript reports several interesting findings, which question
existing models of RNA decay in mammals and provide starting points for further
investigations.

The authors purport that two 'novel' observations were gained from their study. The first - the
contribution of SMG7 to 3'-5' decay to NMD - was observed only when all alternative (major) decay
activities were depleted from the cell, and likely represents a by-pass pathway that only occurs under

these non-physiological conditions. No evidence is provided to suggest that this pathway contributes
to the turnover of NMD substrates under normal conditions.

- • In order to focus on methodological aspects of the xrFrag analysis, we have restructured the
paragraph about the turnover of NMD substrates. In the revised version of the manuscript,
we use this part to demonstrate the combination of xrFrag analysis with the tethering assay
and the knockdowns of decay factors.

The second - the characterization of novel cleavage sites in cytokine 3' UTRs - led the authors to
suggest that cleavage was not mediated by Regnase-1 and that an alternative mechanism is in play
for the decay of TNFalpha mRNA. While this initial observation is intriguing, it is premature (based on
the limited evidence) and the authors should be cautioned to question published findings. Indeed,
the burden of proof lies with these authors to fully characterize these fragments and the mechanism
for the decay of this mRNA before jumping to conclusions.

- • We agree that the results only provide an incomplete analysis of a potentially new and
unreported degradation mechanism, which nevertheless will allow our colleagues to validate
these results with their own approaches. We have revised the manuscript in order to achieve
a balanced presentation of the data concerning IL6 and TNF-alpha, highlighting the
premature nature of our observations. Within the context of a method-oriented manuscript,
it seems unreasonable to elucidate the complete mechanism underlying the endocleavage of
IL6 and TNF-alpha. Keeping in mind that this characterization involves identification of the
endonuclease and all the molecular requirements for cleavage, we feel this would warrant a
comprehensive analysis independently of this study. Moreover, there are other examples of
important decay mechanisms that are still incompletely characterized, e.g. no-go-decay and
non-stop-decay, as for both the endonuclease is still unknown and the mechanism elusive
too. (reviewed in Shoemaker and Green, 2012 Nat. Struct. Mol. Biol.)

2. It needs to be re-iterated that quantitative conclusions cannot be drawn from monitoring ratios of
xrFrag to full-length RNA abundance. (Notably, his measurement is also not 'semi-quantitative', as
suggested by the authors).

- • We thank the reviewer for emphasizing this important point. Please note that we do not
claim that our measurements allow the estimation of absolute numbers of mRNA molecules,
as indicated in all figures ("relative mRNA levels"). The ratios of quantified RNA abundances
are only calculated and shown in order to compare different conditions. However, to avoid
misinterpretation of results obtained by the xrFrag assay, we have revised the discussion of
the manuscript and carefully explain the advantages and disadvantages, as well as potential
applications of the xrFrag method. We also avoid to use the term "semi-quantitative" as it
may be misunderstood by some readers.

This is particularly a concern considering new data indicating that there is a considerable amount of
'leakiness' in the ability of xrRNA elements to block XRN1 and that the extent of the block to XRN1 by
these elements likely differs from construct to construct (as mRNA sequence elements will likely
influence the formation and/or stability of the xrRNA element and its ability to impede XRN1
progression).

- • This is an important point, since the xrRNA has to be compatible with different constructs.
Before we will explain how the problem of leakiness was addressed in the revised version of
the manuscript, it seems important to elaborate on the mechanism of "leakiness" in the
context of xrFrag analysis. Our new data show that xrFrag as well as 3' fragments undergo

slow 5'-3' decay, which results in an apparent leakiness if combinations of two or more
xrRNAs and/or 3' fragments are analyzed. Overall, the stability of the xrFrag is comparable
to 3' fragments, which are widely used to study mRNA turnover pathways. This new
information is not only important for the application of the xrFrag method, but may also help
to better understand the mechanism of xrRNA inhibition during viral infection.

• Although the problem of leakiness was solved as described above, the following aspects
were considered in the original design of the study:

1) In each of the experiments almost identical reporter transcripts were used, which
varied only in small sequence elements. Furthermore, appropriate control reporter
mRNAs were always included and calculated ratios of xrRNA/full length RNAs were
only compared in individual experiments.

2) RNA sequences (50 nt spacer upstream and the 400 nt probe binding sites
downstream) immediately surrounding the xrRNA element were kept consistent.
These sequences serve to exclude or minimize potential effects of surrounding RNA
sequences on folding or efficiency of xrRNA.

3) Except for the stop codon mutation, the TPI/globin NMD reporters have identical
sequences upstream of the xrRNA element. The potential influence of altered mRNP
composition due to the premature translation termination cannot be responsible for
XRN1 blockage, as we see an increase in xrFrag in the case of PTC reporters, which
generally contain more proteins bound to the RNA upstream of the xrRNA (e.g. UPF1,
see Hurt et al. 2013 Genome Res; Zund et al. 2013 Nat Struct Mol Biol).

• To address if mRNA sequence elements can influence the formation and/or stability of the
xrRNA, we constructed novel reporters containing various short RNA sequences upstream of
the xrRNA. We introduced (a) 60 bp long stretches from the RAB7A 3' UTR with different GC
contents (30%; 50%; 70%), (b) 4MS2 binding site as potential XRN1 inhibitors (as shown in
yeast: Garcia and Parker 2015 RNA), or (c) the strong stem loop structure used in other
reporters to block translation. None of these reporter mRNAs produced additional bands,
which would indicate that XRN1 stalls upstream of the xrRNA. Furthermore, the xrRNA was
fully functional in all reporter mRNAs and the full length/xrFrag ratio differed by less than
1.5, which is likely in the range of the normal experimental variability (revised Figure 1f). This
further highlights the versatile usage of the xrFrag system, which even performs accurately
when varying (GC content) or even extremely challenging (secondary structures) RNA
sequences are present.

This reviewer is in complete disagreement that xrFrag abundance offers a superior approach for
studying mRNA turnover over the gold standard of measuring mRNA stability of full-length mRNA. It
does, however, provide a complementary approach that should be of interest to those in the field.

• We agree with the reviewer that our approach complements the existing methods to study
mRNA turnover and will be of interest for researchers in the RNA decay field. The combined
detection of full length mRNAs and xrFrag is a simple but effective approach to initially
characterize robust mRNA degradation mechanisms by giving immediate insight into the
participation of 5'-3' decay. This is of particular interest for future studies interrogating both
novel and existing RNA decay pathways.

• The xrFrag system does not only allow to determine the half-lives of full length mRNAs, but
also potentially allows time-resolved interpretation of the decay directionality as discussed

below. Since the measurement of full length mRNA levels is unaltered by the addition of
xrRNA, the analysis of xrFrag always adds another layer of insight into RNA decay.

- • Nevertheless, we carefully reevaluated all statements made about the advantages of the
xrFrag system in the manuscript and avoid conclusions that may be misinterpreted by the
readers.

Other comments

It is unclear why, in several figures, the abundance of the xrFrag is not significantly reduced after
siRNA knockdown of XRN1 (e.g. Figs. 2b, 3c, 5c, Supp Figs. 2b, 3e). This would seem to indicate that
the xrRNA element blocks other exonucleases in addition to XRN1.

- • xrFrag may potentially be produced by nuclear 5'-3' degradation events, catalyzed for
example by XRN2. To test this hypothesis, we performed subcellular fractionation assays in
order to detect potential nuclear xrFrag (revised Supplementary Figure 1e). In cell lines
expressing the PTC-containing globin mRNA, mainly cytoplasmic xrFrag were detected,
which was expected due to cytoplasmic degradation of the reporter via NMD. The observed
small amounts of nuclear xrFrag may be generated by XRN2, but could also represent a
contamination of the nuclear fractions by small amounts of cytoplasmic RNAs. To further
investigate this, we performed knockdowns isolated or combined knockdowns of XRN1 and
XRN2, however we could not find evidence for XRN2 involvement in xrFrag generation of PTC
mRNAs (data not shown). Considering the new data concerning the stability of xrFrag and 3'
fragments in XRN1-depleted cells, it is apparent that xrFrag do not irreversibly block, but
rather slow down XRN1. The reason why xrFrag levels are not significantly reduced after
XRN1 knockdown is now also explained in the manuscript.

Measurement of xrFrag half-lives upon treatment with ActD is deceiving since steady-state fragment
abundance is a consequence of its production from the turnover of full-length mRNA and the kinetics
of degradation of the fragment itself. Notably, xrFrag production is not inhibited until its precursor,
the full-length RNA, is sufficiently depleted in the cell. Thus, any half-life value calculated from this
approach represents an overestimate of the stability of the RNA fragment and of little value.

- • It should be noted that, although abundances of xrFrag are shown in the manuscript, half-
lives for xrFrag were not measured or directly indicated (Figure 2h). Nevertheless, the
qualitative analysis of xrFrag over time could give more insight in the decay mechanism of
the reporter mRNA. As the reviewer points out, in the case of NMD reporters, the levels of
xrFrag seem to only decrease after almost complete degradation of the substrate transcript.
This indicates that most of the degradation of this mRNA is achieved via the 5'-3' decay
pathway, for which the xrFrag serves as a readout. In contrast, transcripts degraded by 3'-5'
mechanisms will exhibit a different pattern of xrFrag decay over time, which should decline
already from the point of transcriptional shutoff. Therefore, and as mentioned above, the
xrFrag system is compatible with half-life measurements and may even be advantageous
compared to the detection of full length mRNA levels alone.

- • One criticism of the initial submission was that deadenylation and 3'-5' decay are not directly
measured and their contributions can only be inferred by this method. Despite the fact that the
authors introduce new data that is provided to address this concern, neither mRNA polyA tail lengths
nor mRNA 3' ends are directly evaluated and, thus, observations related to these processes are
indirect and conclusions still only inferred.

• It was never claimed in the manuscript that the xrFrag system can directly measure poly(A)
tail lengths or can detect 3'-5' decay and we agree that deadenylation and 3'-5' decay is only
inferred by our method. However, it is established that the data obtained after e.g.
inactivation of the major deadenylase complex CCR4-NOT allows to draw conclusions about
the involvement of deadenylation in the decay of the particular reporter mRNA (see for
example Sandler et al. 2011 Nucleic Acid Res). Furthermore, direct measurements of poly(A)
tail lengths requires different methods, which are to our knowledge fully compatible with the
xrFrag approach. However, as indicated even in the first draft of the manuscript, more time
will be required to improve the technology to this level. This limitation is discussed in the
revised version of the manuscript

• It is unclear why the addition of one or two additional xrRNA elements actually appears to decrease
the efficiency of the most 5' element to block XRN1 activity (note the reduction in xrFragA levels in
Fig 1e). This may indicate that the additional elements either lead to decay of the reporter mRNA to
alternative (non-5'-3') pathways or that these elements can alter the formation/stability of the most
5' element (as suggested above).

• As shown in the manuscript, an optimal readout is achieved when one complete xrRNA
element is used. This recommendation is included in the manuscript.

• The use of two or more xrRNAs in a row is not advantageous, because longer xrFragA are
slowly converted into the shorter forms. The mechanism underlying this observation is
described in the revised version of the manuscript.

Reviewers' Comments:

Reviewer #2 (Remarks to the Author)

The authors have done a nice job to revise their work to emphasize the utility of xrRNA sequences in evaluating mRNA decay pathways in mammalian cells.

The work is clear and convincing, and the analysis of a wide range of RNA decay-inducing events is laudable.

There are a number of changes to the text that are recommended.

Line 42. The use of XRN1-resistant sequences to monitor mRNA decay is not a 'novel molecular approach'. PolyG elements that form G-quadruplexes also serve to inhibit XRN1-mediated 5' – 3' degradation and have been used extensively to interrogate mRNA decay pathways in eukaryotic cells (albeit their utility has been limited to budding yeast). Also see line 419.

Line 86. Based on the presented data, xrRNA elements do not 'halt' XRN1 activity.

Line 98. Steady state RNA levels do not 'demonstrate' enhanced turnover. Kinetic events cannot be shown through RNA abundance measurements. See also Line 188, 194, 195, 258.

Line 101. The xrRNA sequences themselves or not 'resistant' to XRN1.

Line 159. The word 'decelerate' suggests that the rate of catalysis has been reduced. Since XRN1 catalytic activity has not been measured, this statement is not accurate. It is more likely that xrRNAs, when present at the extreme 5' end of an RNA, prevent XRN1 from interacting with the RNA – i.e. they are no longer efficient/preferred substrates for XRN1 activity.

Line 178. "IRES-mediated activation of NMD". IRES do not mediate NMD; however, IRES-mediated translation allows for triggering of NMD from this substrate when the PTC is encountered.

While it is appreciated that the authors consider the decay kinetics of the xrFragments, this material would be better presented as Supplemental Data. In contrast, the data in Supplemental Figure 4 demonstrating the contribution of endo-cleavage versus decapping in NMD should be in the main body of the manuscript.

Figure 4E. When comparing lane 2 and 4, it does appear that knockdown of XRN1 does cause an increase in full-length reporter mRNA levels – even when SMG6 is present in the cell.

Lines 195-199. Residual enzyme activities upon siRNA-mediated knockdown could contribute to the failure of full-length PTC-containing reporter mRNA to reach wild-type levels (Fig. 4e lanes 11&12). This possibility should be included.

Line 343. This may be misleading. What is the evidence that XRN1 remains associated with xrFragments (and thus could be titrated from contributing to the decay of other RNAs)? This should be clarified.

Lines 352-355. The explanation as to why one xrRNA is better than two or three is confusing. How do xrRNAs downstream of the most 5' proximal element influence its efficiency such that xrFragments from that element are less abundant than when only a single xrRNA is present in the transcript?

Supplemental Figure 5e. The authors suggest that a change in migration of xrFragments is indicative of

changes in the polyA tail length and that deadenylation of the mRNA has occurred. While this is not unlikely, the authors can definitively demonstrate polyA tail status by performing RNaseH/oligo(dT) assays to demonstrate which xrFragments are polyadenylated and which are not. This should also be suggested in the Discussion lines 356-363.

REVIEWERS' COMMENTS:

Reviewer #2 (Remarks to the Author):

The authors have done a nice job to revise their work to emphasize the utility of xrRNA sequences in evaluating mRNA decay pathways in mammalian cells.

The work is clear and convincing, and the analysis of a wide range of RNA decay-inducing events is laudable.

There are a number of changes to the text that are recommended.

Line 42. The use of XRN1-resistant sequences to monitor mRNA decay is not a 'novel molecular approach'. PolyG elements that form G-quadruplexes also serve to inhibit XRN1-mediated 5' – 3' degradation and have been used extensively to interrogate mRNA decay pathways in eukaryotic cells (albeit their utility has been limited to budding yeast). Also see line 419.

We have rephrased the sentence in line 72 (not 42): "In this study, we aimed to understand the degradation of different classes of intrinsically unstable mRNAs in mammalian cells using a virus-derived RNA sequence.". Furthermore, we have replaced the word "novel" in line 416 by "powerful".

Line 86. Based on the presented data, xrRNA elements do not 'halt' XRN1 activity.

We have rephrased the sentence "...which were previously reported to block the processively degrading XRN1 upstream of the xrRNA structure²³"

Line 98. Steady state RNA levels do not 'demonstrate' enhanced turnover. Kinetic events cannot be shown through RNA abundance measurements. See also Line 188, 194, 195, 258.

These sentences were changed as follows:

98: "Introducing an NMD-activating PTC (PTC160) in the TPI ORF resulted in decreased steady state reporter levels as well as increased xrFrag abundance, in line with the enhanced turnover of this mRNA (Fig. 1d, lane 2)."

188: "Although we readily detect 3' fragments as a result of endocleavage, knockdown of XRN1 alone lead to only marginally increased levels of the full-length reporter RNA (Fig. 4e and Supplementary Fig. 3c,d)."

194: "Surprisingly, also the combined SMG6+EDC4+DCP2 knockdown could not fully increase the PTC reporter to WT levels, even though elevated PTC reporter levels were observed."

195: *“Interestingly, the knockdown of UPF1 increased the reporter RNAs more than the combined SMG6+XRN1 or SMG6+EDC4+DCP2 depletion, suggesting that not all degradation during NMD can be attributed to decapping or endocleavage.”*

258: *“However, calculating the xrFrag:reporter ratios indicated that both AREs enhanced 5'-3' decay.”*

Line 101. The xrRNA sequences themselves or not 'resistant' to XRN1.

We have replaced the word “sequences” by “structures”

Line 159. The word 'decelerate' suggests that the rate of catalysis has been reduced. Since XRN1 catalytic activity has not been measured, this statement is not accurate. It is more likely that xrRNAs, when present at the extreme 5' end of an RNA, prevent XRN1 from interacting with the RNA – i.e. they are no longer efficient/preferred substrates for XRN1 activity.

We have rephrased the sentence: “This finding suggests that in mammalian cells xrRNAs do not irreversibly trap, but impair XRN1 activity to such an extent that xrFragB can be detected.”

Line 178. “IRES-mediated activation of NMD”. IRES do not mediate NMD; however, IRES-mediated translation allows for triggering of NMD from this substrate when the PTC is encountered.

We have rephrased the sentence: “The activation of NMD by IRES-mediated translation will either lead to endocleavage or decapping and the accumulation of xrFragB or xrFragA, respectively.”

While it is appreciated that the authors consider the decay kinetics of the xrFrag, this material would be better presented as Supplemental Data. In contrast, the data in Supplemental Figure 4 demonstrating the contribution of endo-cleavage versus decapping in NMD should be in the main body of the manuscript.

The kinetic analysis of the xrFrag will be important for the application of the method by other scientists (e.g. when would be an optimal time point to harvest RNA). In contrast, the contribution of endocleavage vs. decapping addresses a biological question. We therefore prefer to show the kinetic data as main figure.

Figure 4E. When comparing lane 2 and 4, it does appear that knockdown of XRN1 does cause an increase in full-length reporter mRNA levels – even when SMG6 is present in the cell.

The less than two-fold decrease of the ratio of xrFrag:reporter from 145 (lane 2) to 88 (lane 4) is shown in the quantification of Figure 4e. We have therefore rewritten the sentence: “Although we readily detect 3' fragments as a result of endocleavage, knockdown of XRN1 alone lead to only marginally increased levels of the full-length reporter RNA (Fig. 4e and Supplementary Fig. 3c,d).”

Lines 195-199. Residual enzyme activities upon siRNA-mediated knockdown could contribute to the failure of full-length PTC-containing reporter mRNA to reach wild-type levels (Fig. 4e lanes 11&12). This possibility should be included.

We have added the sentence: “However, we cannot exclude the possibility that the incomplete depletion of the decay factors contributes to the observed discrepancy.”

Line 343. This may be misleading. What is the evidence that XRN1 remains associated with xrFrag (and thus could be titrated from contributing to the decay of other RNAs)? This should be clarified.

To clarify, we have rephrased the sentence: “Although high levels of xrFrag could potentially inhibit cellular 5’-3’ decay due to the sequestration or inactivation of XRN1, as suggested previously⁴⁰, we did not observe signs of such an inhibition.”

Lines 352-355. The explanation as to why one xrRNA is better than two or three is confusing. How do xrRNAs downstream of the most 5’ proximal element influence its efficiency such that xrFrag from that element are less abundant than when only a single xrRNA is present in the transcript?

It is correct, that the downstream xrRNA will not influence the efficiency of the 5’ proximal xrRNA. However, all xrRNAs will appear as additional xrFrag bands on the northern blot, which will make the analysis less clear. To clarify, we have revised the sentence: “Interestingly, for exact xrFrag analysis a single xrRNA is superior to a combination of two or three xrRNAs, which do not significantly elevate the amount of the 5’ xrFrag, but lead to the accumulation of additional xrFrag.”

Supplemental Figure 5e. The authors suggest that a change in migration of xrFrag is indicative of changes in the polyA tail length and that deadenylation of the mRNA has occurred. While this is not unlikely, the authors can definitively demonstrate polyA tail status by performing RNaseH/oligo(dT) assays to demonstrate which xrFrag are polyadenylated and which are not. This should also be suggested in the Discussion lines 356-363.

This point is now discussed and a new reference included: “Although some of our results indicate that xrFrag allow to assess poly(A) tail lengths, specific assays such as RNaseH-oligo(dT) digestion or PAT assays are needed to improve this aspect of our experimental system.”